# Description and validation of VERT 1.0, an R-based framework for estimating road transport emissions from traffic flows

Giorgio Veratti[1], Alessandro Bigi[1], Sergio Teggi[1], and Grazia Ghermandi[1]

[1]Department of Engineering 'Enzo Ferrari'. University of Modena and Reggio Emilia, 41125 Modena, Italy

**Correspondence:** Giorgio Veratti (giorgio.veratti@unimore.it)

**Abstract.** VERT is an R package developed to estimate traffic emissions of a wide range of pollutants and greenhouse gases based on traffic estimates and vehicle fleet composition data, following the EMEP/EEA methodology. Compared to other tools available in the literature, VERT is characterised by ease of use and rapid configuration, while maintaining great flexibility in user input. It is capable of estimating exhaust, non-exhaust, resuspension, and evaporative emissions and is designed to

accommodate future updates of available emission factors. In this paper, case studies conducted at both urban and regional scales demonstrate VERT's ability to accurately assess transport emissions. In an urban setting, VERT is integrated with the Lagrangian dispersion model GRAMM-GRAL and provides $NO_x$ concentrations in line with observed trends at monitoring stations, especially near traffic hotspots. On a regional scale, VERT simulations provide emission estimates that are highly consistent with the reference inventories for the Emilia-Romagna region (Italy). These findings make VERT a valuable tool for

air quality management and traffic emission scenarios assessment.

## 1   Introduction

The provision of clean air is recognized as a fundamental necessity for human health and general well-being. However, the World Health Organization (WHO) estimates that almost all of the world's population (99%) breathes air that exceeds the recommendations proposed in the latest air quality guidelines (WHO, 2021), with low- and middle-income countries suffering

from the highest exposures. In Europe, for example, despite significant reductions in emissions and ambient concentrations over the past decade, a staggering 97% of the urban population is still exposed to particulate matter ($PM_{2.5}$) concentrations above $5\,\mu g\,m^{-3}$, a threshold set forth by WHO to protect public health (WHO, 2021).

Given the compelling scientific evidence of the severe health effects associated with ambient air pollution (Fuller et al., 2022; Song et al., 2022; Yolton et al., 2019; Landrigan et al., 2018; Costa et al., 2017; Loomis et al., 2013), accurate estimation of

pollutant concentrations and related emissions is essential for developing effective mitigation strategies. To accomplish this task, numerous international agencies, research institutions, and environmental organisations are collecting data to identify and monitor emissions worldwide. Their objective is to compile specific emission inventories using a standardised and transparent process that can be regularly updated over time through a consistent approach.

Emission inventories are generally classified as either bottom-up or top-down, depending on the estimation approach used.

While both methods quantify total emissions through the product of emission factors and activity indicators, the top-down

approach aggregates activity data at a large scale, such as national or regional, before allocating emissions to sub-areas based on activity dependent patterns. Conversely, the bottom-up approach estimates emissions for individual activities and then aggregates them at the spatial resolution required for a specific application. The top-down approach is generally preferred for large-scale inventories where the identification of individual activities may be impractical due to a lack of site-specific data or time-consuming computations. Bottom-up inventories, on the other hand, are preferred when detailed and reliable information on activity indicators is available, despite the more time-consuming nature of emission estimation. Examples of top-down emission inventories are HTAP_v3 (Crippa et al., 2023), TNO-MACCII and MACCIII (Kuenen et al., 2014), E-PRTR and JRC07 (Trombetti et al., 2018), while examples of bottom-up based inventories are EDGAR (Janssens-Maenhout et al., 2019) or regional and national emission inventory that generally cover limited areas, such as INEMAR (Marongiu et al., 2022; INEMAR, 2019) or the Italian national emission inventory (ISPRA, 2019). Other catalogues, such as CAMS-REG (Kuenen et al., 2022) and EMEP (Ullrich et al., 2023), effectively use the strengths of both top-down and bottom-up approaches, resulting in hybrid inventories that provide a comprehensive and reliable representation of emissions at the continental scale.

Emissions from the transport sector currently stand out as a significant source of anthropogenic pollutants in many urban areas of the world (Hooftman et al., 2018; Jonson et al., 2017; Squires et al., 2020; Degraeuwe et al., 2017; Veratti et al., 2023; Ghermandi et al., 2020, 2019). Combustion processes in vehicle engines contribute to the release of several air pollutants, including both primary particulate matter (PM) and other gaseous compounds, such as nitrogen oxides ($NO_x$), volatile organic compounds (VOCs), ammonia ($NH_3$), and sulphur dioxide ($SO_2$), which are important precursors for the formation of secondary particulate matter and photochemical smog (Moussiopoulos et al., 1995; Nogueira et al., 2015; Jeong et al., 2019; Karagulian et al., 2015; Hao et al., 2020). In addition to the exhaust component, traffic-related non-exhaust emissions, including tyre, road, and brake wear, as well as resuspension, contribute significantly to total PM concentrations measured in urban environments. Recent estimates from different countries indicate that the non-exhaust fraction accounts for 60% to 90% of PM10 and 25% to 85% of PM2.5 from road traffic emissions (Piscitello et al., 2021). While policymakers in regions such as Europe, US and China, are making efforts to promote vehicle electrification, transport remains a significant source of non-exhaust emissions, which are becoming increasingly important as vehicle mass increases (Beddows and Harrison, 2021; Piscitello et al., 2021; Liu et al., 2022). Therefore, the need for a comprehensive understanding and accurate estimation of vehicle emissions remains crucial.

Various emission models have emerged in the past decade to evaluate the impact of traffic on atmospheric emissions. Examples include the Motor Vehicle Emissions Simulator (MOVES) by the U.S. Environmental Protection Agency (Yao et al., 2014), the High-Elective Resolution Modelling Emission System (HERMES, Guevara et al., 2019, 2020) developed by the Barcelona Supercomputing Center, TREFIC by ARIANET S.r.l. (Pallavidino et al., 2014; Crosignani et al., 2021; Fabbi et al., 2022), Vehicular Emissions INventory (VEIN) by Ibarra-Espinosa et al. (2018), CARS (Comprehensive Automobile Research System) by Baek et al. (2022), and Yeti, a traffic emission inventory framework based on the Handbook Emission Factor for Road Transport (HBEFA, Chan et al., 2023). However, despite the progress made, none of these models can fully meet the diverse needs of environmental experts, modellers, and policy makers due to their inherent strengths and limitations. These tools are tailored to specific user needs and use different development approaches. The characteristics of each model depend

on factors such as the type of traffic activity, the method used to calculate emissions, the distribution of vehicle speeds and the geographical resolution of inputs and outputs. As a result, each tool has its own level of specificity based on the different modelling assumptions included in its framework. The choice of a particular model therefore depends on the objectives of the study or application.

The major limitations of current transport emissions models concern their adaptability to scenarios different from those in which they were developed. An example is MOVES, which faces complexities when applied beyond U.S. borders. Accessibility is further hampered by certain models, like TREFIC, which require a proprietary licence that limits their use. In addition, alternatives such as VEIN and HERMES require both time-consuming operational procedures and technical skills to generate new case studies based on local data, creating practical barriers to their seamless implementation.

In this study, we present VERT (Vehicular Emissions from Road Traffic), a transport emissions modelling tool developed in the R programming language. It is specifically designed to estimate traffic emissions using a simple and user-friendly framework to facilitate its use by individuals with limited programming skills. Aligned with the EMEP/EEA air pollutant emission inventory guidebooks (Ntziachristos and Samaras, 2023; Ntziachristos and Boulter, 2023; Mellios and Ntziachristos, 2023) and consistent with the 2006 IPCC guidelines for greenhouse gas emissions, VERT requires two key inputs, the local fleet composition

and an estimate of traffic flows along the target road network. The model has been structured to handle traffic information with different levels of detail, since these depend on the traffic data availability, ensuring significant adaptability to different case studies while maintaining user-friendly applicability.

In the first part of the study, VERT is introduced and its implementation methodology is described. In the second section, VERT is coupled with the Lagrangian modelling system GRAMM-GRAL (Oettl, 2015a, b, c) to assess $NO_x$ concentrations in

an urban hot spot of the Po valley (Italy). Then, in the third section, VERT is applied on a broader area, covering the Emilia-Romagna region of the Po valley, to estimate traffic emissions from a larger road infrastructure comprising approximately 7,000 streets. This latter estimate is further validated comparing VERT results with the most recent and up-to-date regulatory emission inventory of the same area. Finally, some conclusions are drawn in the last section.

## 2   VERT description

The main goal of VERT is to estimate transport-related emissions using a bottom-up approach following the EMEP/EEA methodology (Ntziachristos and Samaras, 2023; Ntziachristos and Boulter, 2023; Mellios and Ntziachristos, 2023). In this framework, activity data is represented by the number of vehicles travelling on a given road segment and the representative emission factor is calculated using information on the local fleet composition, vehicle speed, meteorological conditions and

topological characteristics of the road segment, such as length and slope.

VERT is capable of estimating emissions for a wide range of pollutants and greenhouse gases, including CO, $NO_x$, Non-Methane VOC (NMVOC), PM, black carbon, organic carbon, $NH_3$, $SO_2$, $N_2O$, $CO_2$, and $CH_4$. While the standard computational framework is structured to evaluate traffic-related emissions at an hourly time step, VERT provides the versatility to

seamlessly adapt to the specific requirements and input characteristics of a given study. More specifically, if the vehicle flow
data provided to VERT reflect a longer time interval, the resulting emissions calculations will be adjusted to the temporal
resolution corresponding to the input provided to the model. This adaptability ensures that the analysis remains consistent and
fit also to input data with time resolution different from the 1-hour standard.

For greater user flexibility, three different types of vehicle flows can be provided to VERT. These options are consistent with
standard estimates derived from well-established macroscopic traffic models (Helbing, 1995; Johari et al., 2021; Heyken Soares
et al., 2021; Krajzewicz, 2010). Specifically, users can choose to input a single cumulative traffic flow that includes all vehicle
categories. Alternatively, for a more detailed analysis, users can enter two different flows, one for light vehicles (such as cars
and mopeds/motorcycles) and another for commercial vehicles (including both light and heavy types). A third option allows the
user to enter four separate flows corresponding to cars, mopeds/motorcycles, light duty trucks and heavy duty trucks. Finally,
the fleet composition required by VERT must be adjusted according to the number of flows selected in input.

For a general road segment denoted as $k$, a general parking lot $m$ and a specific pollutant denoted as $i$, the on-road emission ($E$)
is calculated based on five components, as outlined in the Eq. (1):

$$E_i^k = E_i^k hot \ + \ E_i^k cold \ + \ E_i^k non-exhaust \ + \ E_i^k resuspesions \ + \ E_i^{k,m} evaporative \tag{1}$$

Here, $E_i^k hot$ represents hot exhaust emissions, $E_i^k cold$ refers to emissions during transient thermal engine operation, com-
monly known as cold-start emissions. $E_i^k non-exhaust$ refers to PM emissions resulting from mechanical parts wear or
caused by road and tyre abrasion. $E_i^k resuspensions$ quantify the amount of PM that was deposited on the road surface and
subsequently resuspended into the atmosphere due to vehicle movement. $E_i^k evaporative$ encompasses emissions of organic
gaseous compounds released into the atmosphere due to tank or running losses. The following subsections of the text outline
the methodologies employed to estimate each of these components.

## 2.1 Hot exhaust emissions

The combustion process in a vehicle engine is a complex series of chemical reactions that occur within the engine's cylinders.
It involves the mixing of fuel and air, followed by ignition, resulting in the release of energy that propels the vehicle. While
stoichiometric complete combustion of hydrocarbon fuels with oxygen ideally produces only $CO_2$ and $H_2O$, the real-world
combustion processes inevitably involve the formation of various pollutants such as carbon monoxide (CO), hydrocarbons
(HC), and PM. These by-products are not fully controlled by the aftertreatment equipment and are consequently released
into the atmosphere. The abundance of nitrogen ($N_2$) and oxygen ($O_2$) in the air mix, along with sulphur compounds in the
fuel, creates additional pollutants, such as $NO_x$ and $SO_x$, that pose additional environmental challenges. Furthermore, while
aftertreatment devices are effective in reducing the emissions of the previously mentioned pollutants, they may also generate
$NH_3$ and $N_2O$ due to inefficiencies in the conversion processes.

Hot exhaust emissions are influenced by a variety of factors. These include vehicle characteristics such as fuel type, engine
size, emission standard, vehicle mileage, load and mass, but also depend on road characteristics like pavement condition, slope

and length. All of these aspects are considered by VERT and integrated into Eq. (2), which is used to estimate hot exhaust emissions (Ntziachristos and Samaras, 2023).

$$E_i^k hot = EF_i hot \cdot EF_i dgr \cdot imp.fuel_i \cdot n.veh^k \cdot L^k \qquad (2)$$

In this formulation, $EF_i hot$ is the hot emission factor, while $n.veh^k$ and $L^k$ are respectively the number of vehicles travelling on the road segment $k$ and the length of the road segment $k$ itself. Two additional factors, $EF_i dgr$ and $imp.fuel_i$, are included in the calculation to correct the baseline $EF_i hot$ for vehicle mileage and fuel characteristics. More specifically, the baseline $EF_i hot$ refers to a fleet with an average mileage between 30,000 and 60,000 km, then the $EF_i dgr$ factor is introduced to correct for the increase in hot emissions resulting from vehicles with higher mileage. As the use of improved fuels has been mandatory in the EU since 2000, the $imp.fuel_i$ coefficient is used to accounts for the reduced emissions due to their use for vehicles older than that year. In Eq. (2), it is of utmost importance to accurately estimate $EF_i hot$, as this factor encompasses vehicle and road characteristics. VERT provides two different alternatives to estimate baseline $EF_i hot$:

1. Speed dependent $EF_i hot$. In this relation, the hot exhaust EF is directly related to the vehicle speed and the corresponding formulation is given in Eq. (3), where *v* is the vehicle speed, while *A*, *B*, *C*, *D*, *E*, *F* and *G* are experimental coefficients derived from tests on real road driving cycles and laboratory tests. These tests have been carried out as part of several scientific projects, including the EUCAR/JRC/CONCAWE programme, the European Commission's PARTICULATES project, the European Commission's ARTEMIS project and the COST 319 action, among others. The experimental coefficients *A*, *B*, *C*, *D*, *E*, *F* and *G* are obtained by regression analysis, resulting in a polynomial curve that fits the observed data and provides a general expression valid for each vehicle category (Kouridis et al., 2010). These coefficients are stored in dedicated data frames in VERT and are used during model execution. They depend on vehicle type, fuel, emission standard, engine size, road characteristics and duty trucks load. By providing the local fleet composition and vehicle speed, VERT internally calculates an average $EF_i hot$ for the given condition. In addition, in order to better reflect emissions in traffic jams or very congested conditions, a correction factor has been introduced to take account of increased emissions at very low speeds. Specifically, when the vehicle speed falls below the threshold of the validity range of the proposed coefficients, the time spent on the road is increased by a factor *w*, calculated as the ratio between the lower speed threshold and the specific speed, down to a minimum limit of 3 $km\ h^{-1}$. This factor reflects the increased emissions observed in various studies such as Zamboni et al. (2015), Lejri et al. (2018) and Lejri and Leclercq (2020). However, it should be noted that the model is tailored to driving scenarios and therefore idling emissions may not be accurately estimated.

$$EF_i hot = (A \cdot v^2 + B \cdot v + C + D/v)/(E \cdot v^2 + F \cdot v + G) \qquad (3)$$

In the formulation proposed in Eq. (3), it is important to emphasise that the VERT model is not designed to estimate vehicle travel speeds; rather, this variable is an input to the model. When only traffic flows are available for a given road,

various empirical flow-speed relations can estimate vehicle speeds based on peak hour traffic flows and the road's vehicle capacity. Examples of these formulations are provided by Brilon and Lohoff (2011); Verhoef (2005) for motorways and by Al-Bahr et al. (2022); Juhász et al. (2016) for urban traffic situations. The outputs of these relations can then be used as input to VERT.

2. EF based on fuel and lubricant consumption. Alternatively, users can enter their specific EF values for fuel and lubricant consumption, expressed as mass of pollutant per mass of fuel or lubricant consumed. In this case, VERT uses fleet composition along with Eq. (2) to estimate total fuel and lubricant consumption ($EF_i hot$ becomes the energy consumption factor), which are then combined with the user input to estimate total emissions.

## 2.2 Cold start emissions

Cold-start emissions refer to the additional release of pollutants by a vehicle's engine during the initial phase of operation, i.e. when the engine itself and the catalytic converter system have not yet reached their optimal operating temperature range. This typically occurs during engine startup, such as when a vehicle starts from a parked location or a residential area. While cold-start emissions can occur in all driving conditions, they are more common in urban and rural driving, because highway starts are comparatively limited. In addition, cold-start events are inherent to all vehicle types, although comprehensive data for accurate estimation are primarily available for gasoline, diesel, and LPG cars, including light-duty trucks. Based on these considerations, VERT accounts for cold-start emissions only for passenger cars and light duty trucks on urban and rural roads, using the Eq. (4):

$$E_i^k cold = EF_i hot \cdot ([EFcold/EFhot]_i - 1) \cdot \beta \cdot n.veh^k \cdot L^k \tag{4}$$

$\beta$ represents the fraction of mileage driven with a cold engine or the catalyst system operating below the light-off temperature, with respect to the mean trip distance. $EF_i hot$ denotes the hot emission factor, and the $[EFcold/EFhot]_i$ ratio is computed using the general expression reported in the Eq. (5), where $H$ and $I$ are empirical coefficients that vary according to the emission standard, engine size and vehicle speed, while $T$ is the mean air temperature for the period of interest. The $\beta$ parameter (Eq. (6)), on the other hand, depends on the average trip distance ($Lt$), defined as the trip segment between a key-on and a key-off event, which can be set as input according to the user's case study.

$$[EFcold/EFhot]_i = H + I \cdot T \tag{5}$$

$$\beta = 0.6474 - 0.02545 \cdot Lt - (0.00974 - 0.000385 \cdot Lt) \cdot T \tag{6}$$

## 2.3 Non-exhaust emissions and resuspension

Non-exhaust emissions encompass various compounds, such as black carbon, organic carbon, metals, ions or more generally PM, which are not directly associated with fuel combustion but instead arise from the wear and tear of vehicle components and

road abrasion. In addressing these emissions, VERT employs a vehicle speed-dependent approach following the Eq. (7):

$$E_i^k non-exhaust = EF_i TSP \cdot Fs \cdot Ss(v) \cdot n.veh^k \cdot L^k \tag{7}$$

The emission factor $EF_i TSP$ represents the total suspended particulate emissions per unit vehicle, which varies by vehicle type. $EF_i TSP$ can be converted to specific fractions of particulate matter (e.g. $PM_{10}$, $PM_{2.5}$, $PM_1$, $PM_{0.1}$) or black carbon by using different values of *Fs*, which acts as a size-scaling factor. In addition, *Ss(v)* serves as a coefficient that adjusts the emission estimate to account for travelling speed. More detailed information on the reference $EF_i TSP$, *Ss(v)*, and *Fs* used for
tire and brake emissions can be found in Ntziachristos and Boulter (2023).

Due to the limited understanding of airborne particle emissions resulting from road surface wear, the methodology for estimating their contribution has not yet reached a level of detail that allows for a refined approach based on travelling speed. Therefore, VERT sets the parameter *Ss(v)* equal to 1 when calculating road surface emissions.

Significant uncertainties also persist in estimating the contribution of resuspended dust aerosols from traffic activities, as re-
200 ported in several studies (Amato et al., 2016; Harrison et al., 2021; Casotti Rienda and Alves, 2021). To address this challenge, VERT provides the user with the flexibility to choose between two calculation methods to ensure adaptability to different case studies. The first approach is based on the EPA-42 methodology published by the U.S. Environmental Protection Agency (EPA, 2011). This formulation includes variables such as the average mass of the circulating fleet (*W*), the surface silt loading of the road (*sL*), a size speciation factor that accounts for the PM mass size distribution, and the frequency of precipitation during the
205 reference period of the simulation (*perc.wet.days*). The calculation follows the Eq. (8).

$$E_i^k resuspensions = (sL^{0.91}) \cdot Fs \cdot (W^{1.02}) \cdot (1 - (1/4 \cdot perc.wet.days)) \cdot L^k \tag{8}$$

Since the latter approach is sometimes considered to overestimate the resuspension component (Pachón et al., 2018; Venkatram, 2000), an alternative option is provided to the user. In this alternative, the user has the flexibility to manually select and enter their own resuspension EF, allowing for customization based on the specific types of vehicle flows being considered in
the calculation. For this option, the default emission factors are in the range of those proposed by Amato et al. (2012).

## 2.4 Evaporative emissions

Evaporative emissions from vehicles refer to the release of volatile gaseous compounds into the atmosphere due to the vaporisation of liquid fuels or other volatile components in the vehicle's fuel system. These emissions consist of three primary components: running losses, diurnal emissions from the tank, and soak emissions. Running losses occur during vehicle opera-
215 tion and involve the evaporation of fuel vapours from the fuel system and engine under normal driving conditions. Conversely, diurnal and soak emissions occur when the vehicle is parked with the engine turned off.

Diurnal emissions result from the increase in ambient temperature, which causes the expansion of fuel vapours in the fuel tank. Despite the presence of emission control canisters in most present-day tanks, the importance of evaporative VOC leaks remains. To quantify the daily emissions from fuel tanks, VERT uses the following formulation (Eq. (9)), where $EF_i diu$ is the

daily emission factor depending on the vehicle category, *n.day* is the number of days considered in the simulation and *n.veh* is the number of vehicles in a given parking lot *m*.

$$E_i^m diurnal \ = EF_i diu \ \cdot \ n.days \cdot \ n.veh^m \tag{9}$$

Soak emissions are quantified using Eq. (10), where *n.trip.day* is the average number of trips per day, $\gamma$ is the fraction of petrol vehicles equipped with carburettors and/or fuel return systems, $EF_{hot,carb}soak$ and $EF_{cold,carb}soak$ are the emission factors for petrol vehicles equipped with carburettors for hot and warm/cold emissions respectively, while $EF_{hot,inj}soak$ is the mean hot soak emission factor for petrol vehicles equipped with fuel injection and fuel returnless systems.

$$E_i^m soak \ = n.days \cdot \ n.veh^m \cdot \ n.trip.days \cdot \ [\gamma \cdot ((1-\beta) \cdot \ EF_{hot,carb}soak \ + \beta \cdot \ EF_{cold,carb}soak \ ) + (1-\gamma) \cdot \ E_{hot,inj}soak \ ] \tag{10}$$

Running losses are expressed as in the Eq. (11), with $\beta$ defined as in the Eq. (6), $EF_{hot,carb}run$ and $EF_{cold,carb}run$ are the related emission factors for petrol vehicles equipped with carburettors for hot and warm/cold emissions and $EF_{hot,inj}run$ is the related emission factor for petrol vehicles equipped with fuel injection and fuel returnless systems:

$$E_i^k run \ = Lt^{-1} \cdot \ n.veh^k \cdot \ L^k \cdot \ [\gamma \cdot \ ((1-\beta) \cdot \ EF_{hot,carb}run \ + \beta \cdot \ EF_{cold,carb}run \ ) + (1-\gamma) \cdot \ EF_{hot,inj}run \ ] \tag{11}$$

Finally, total evaporative emissions are calculated as the sum of diurnal, soak and running emissions, see Eq. (12):

$$E_i^{k,m}evaporative \ = E_i^m diurnal \ + E_i^m soak \ + E_i^k run \tag{12}$$

## 2.5 VERT configuration and design

VERT is an open-source traffic emission model developed in the R programming language. It acts as an user-friendly framework, making it easy for those with basic programming skills to assess emissions on a reference road network. The model is intentionally designed for simplicity and has no mandatory dependencies on other packages, ensuring seamless out-of-the-box functionality and high portability across different operating systems and machines. While the VERT model is self-contained, it also allows for greater flexibility in both computation and data processing through the ability to integrate external packages. This flexibility is especially beneficial when users want to speed up computations on large road network datasets. For example, VERT includes a feature that allows for "embarrassingly" parallelization, i.e. the separation of the computation in a number of independent parallel tasks, through the "parallel" package. In this approach, instead of sequentially looping through each segment of the road network, the computation of all road segments is distributed to different cores of the machine, resulting in a significant reduction in overall computation time.

For added convenience in data preparation, input management or results visualisation, users can also choose to integrate add-on packages such as "sf", "dplyr" and "ggplot2". These further enhance the overall user experience by providing tools for streamlined operations and insightful visual representation of the final output, although they are not strictly necessary for the emissions calculation. Section S1 in the Supplementary material provides vignette documentation tailored to assist users in estimating emissions for a specific road network using sample inputs. This guide also emphasises how the features of external

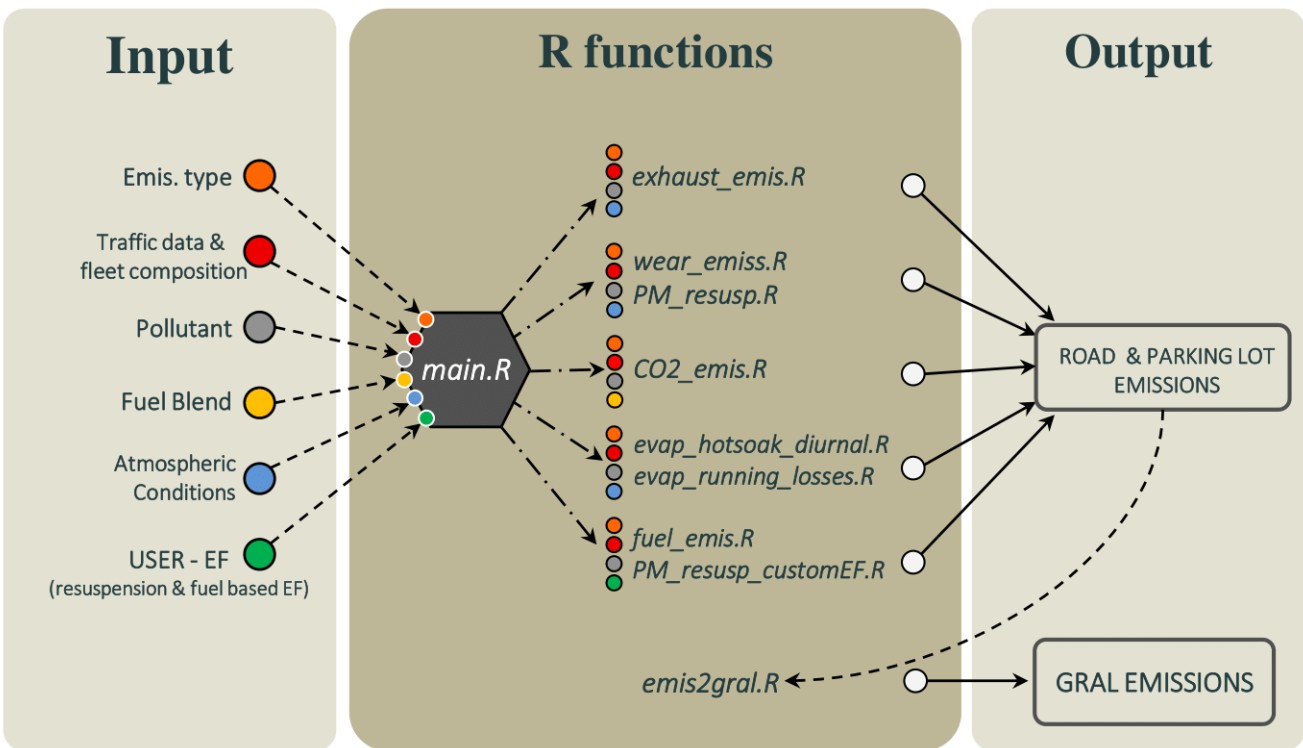

**Figure 1.** Schematic representation of the VERT structure.

packages can be used to extend the capabilities of VERT, covering aspects such as input data arrangement, computation, and result visualisation. User manual documentation is also provided along with each function and data implemented in the R package.

Fig. 1 shows a schematic representation of the VERT structure along with the execution workflow. The left side of the figure shows the primary inputs that are critical to the model. These include the emission calculation method, the pollutant of interest, traffic flow data, fleet composition, fuel blend composition, and atmospheric conditions. In addition, if the user chooses to estimate exhaust or resuspension emissions using user-defined EF, these must be provided as input to VERT.

Although each VERT utility can be called individually, the simplest and most widely used method is to pass all input parameters to the *main.R* function. This streamlined approach effectively manages the emissions calculation by triggering the necessary utilities based on user specifications. The output of *main.R* is then stored directly in the attribute table of the road network spatial features, accurately assigning emissions to each road segment and facilitating post-processing procedures. This defined structure is also suitable for traffic emission input files for dispersion models such as GRAMM-GRAL, for which a dedicated function, *emis2gral.R*, has been developed.

As periodic updates of emission factors are continuously made available, VERT has been designed to include them within its framework. In the current release, two sets of EF are available for calculation, corresponding to the 2020 and 2023 publications

for hot and wear EF, with the latter including updates for Euro 6 light-duty vehicles and Euro VI heavy-duty vehicles. Additionally, since the estimation of PM speciation into black carbon and organic carbon from traffic remains subject to significant uncertainties (Lugon et al., 2021; Flores et al., 2020; Markiewicz et al., 2017; Tian et al., 2021), VERT also provides the option to generate emission estimates for these two components with the uncertainty defined in the EEA guideline (Ntziachristos and Samaras, 2023; Ntziachristos and Boulter, 2023). Users can choose to run the calculation based on the suggested speciation factor, or with the lower or upper uncertainty thresholds, allowing users to tailor the output of VERT to their specific needs and preferences.

## 2.6 Computation performances

The performance of VERT in computing emissions was tested on different machines, including a 2-Core laptop (Intel i7-5500U 2.40GHz), a 16-Core server (AMD EPYC 7313P 3.0 GHz), a 20-Core cluster node (Intel Xeon Gold 6230 2.10GHz), and a 52-Core cluster node (Intel Xeon Gold 5320 2.20GHz). Tests were performed on each machine with a progressively increasing number of cores, starting from one up to the maximum number available, with the number of cores doubling in each successive run. For each test, a road network consisting of 500 streets in the urban area of Modena (a sub-sample of the road network shown in Fig. 3) was used. Hourly emissions for the morning rush hour were calculated for the following pollutants and greenhouse gases: CO, VOC, $NO_x$, $CH_4$, $CO_2$, PM exhaust, BC exhaust, OC exhaust, $SO_2$, $NH_3$, $N_2O$, brake wear, surface wear and tyre wear for all three PM sizes (TSP, $PM_{10}$, $PM_{2.5}$), evaporative VOC and resuspension using Eq. 8.

Fig. 2 shows the computing time for different machines and core configurations. The results indicate that for each machine tested, the computation time decreases as the number of cores increases, with an almost proportional improvement, i.e. doubling the number of cores roughly halves the computation time. For example, on the AMD EPYC 7313P 3.0 GHz machine, processing the same 500 street sample takes 8,256 s with a single core, 4,171 s with two cores, 2,101 s with four cores, 1,122 s with eight cores, and 620 s with sixteen cores. Similar performance improvements were seen on the other machines tested.

It is important to note that the computational cost of VERT increases proportionally with the number of segments included in the reference road network, regardless of the geometric complexity and detail of the road network. For example, the calculation of emissions for a road segment with homogeneous characteristics (such as traffic flow, driving speed, road gradient and silt load) will take significantly less time than for a segment of the same length and spatial resolution but with varying characteristics, where any variation in traffic flow, speed, gradient and silt load will increase the computational load.

## 3 Case Study 1 - VERT for urban scale dispersion modelling

Modena, shown in Fig. 3, is the focus of the first case study. The city is located in the southern region of the Po valley at an elevation of approximately 35 metres above sea level and has a population of approximately 180,000 inhabitants. The Po valley, in which Modena is located, is a flat plain bordered by the Alps to the north and the west, and by the Apennines to the south. This topographical arrangement has a significant impact on the local climate, influencing weather patterns and potentially trapping low level air masses within its natural boundaries. In particular, the valley often suffers from low wind conditions, preventing

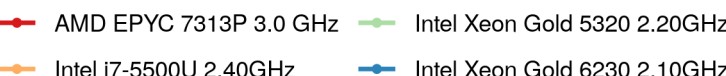

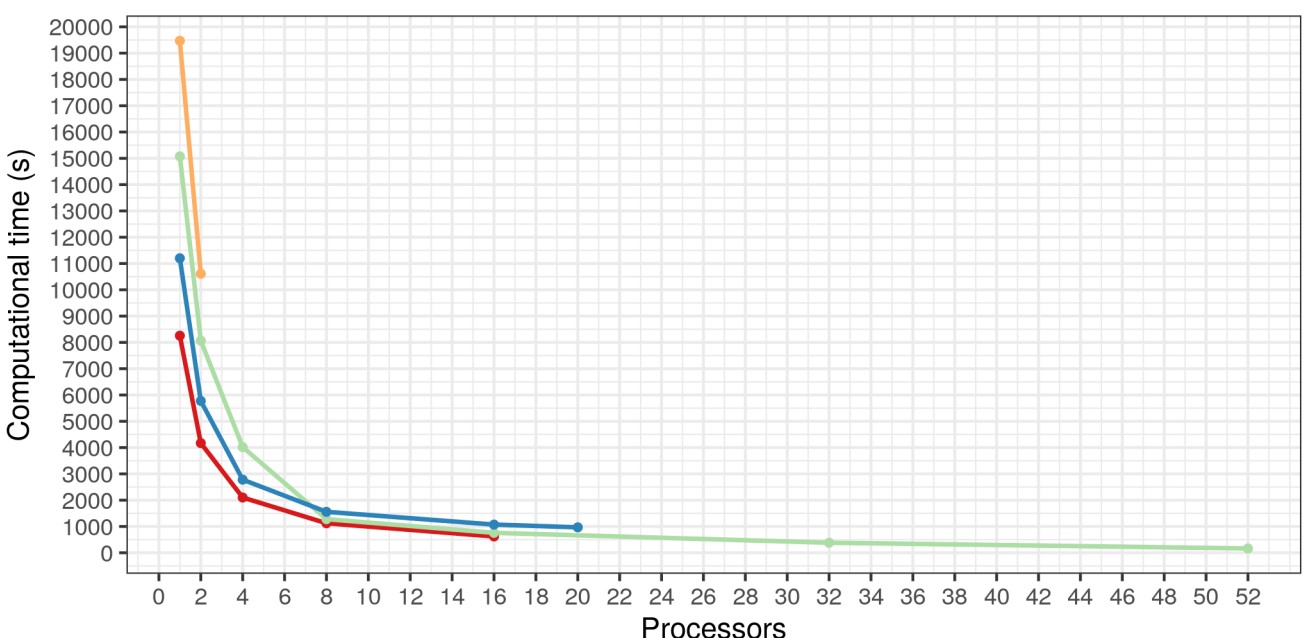

**Figure 2.** Computational times (seconds) from the scalability test using the VERT package applied to a road network of 500 streets on four different machines.

the effective dispersion of ground emissions and contributing to the accumulation of pollutants. This is further exacerbated in the winter months by atmospheric inversion, which reduces the extent of vertical mixing and thus the part of the atmosphere where pollutants are diluted and mixed (Bigi et al., 2012, 2023; Pernigotti et al., 2012). These meteorological characteristics,

together with the high population density and the presence of busy commercial and industrial activities, places Modena among the largest European cities that exceed the air quality limits set by both the European regulation (European Council, 2008) and the latest WHO guidelines (WHO, 2021).

The emission inventory for the Emilia-Romagna region (INEMAR, 2019) estimates that vehicular traffic serves as the predominant source of $NO_x$ emissions in Modena, contributing with 78% to the total emissions, followed by domestic heating

(12%), other mobile machinery (3%), waste treatment management (3%) and the industrial sector (2%). Previous studies (Bigi et al., 2023; Veratti et al., 2021, 2020a, b, 2017) have evaluated the impact of different sources on air quality in the city and its surrounding areas. This paper, however, focuses specifically on transport activities within the city and examines the influence of traffic on urban air quality using an integrated modelling approach.

In the following subsection, the integrated modelling approach is described, followed by its application to a real-world case

study.

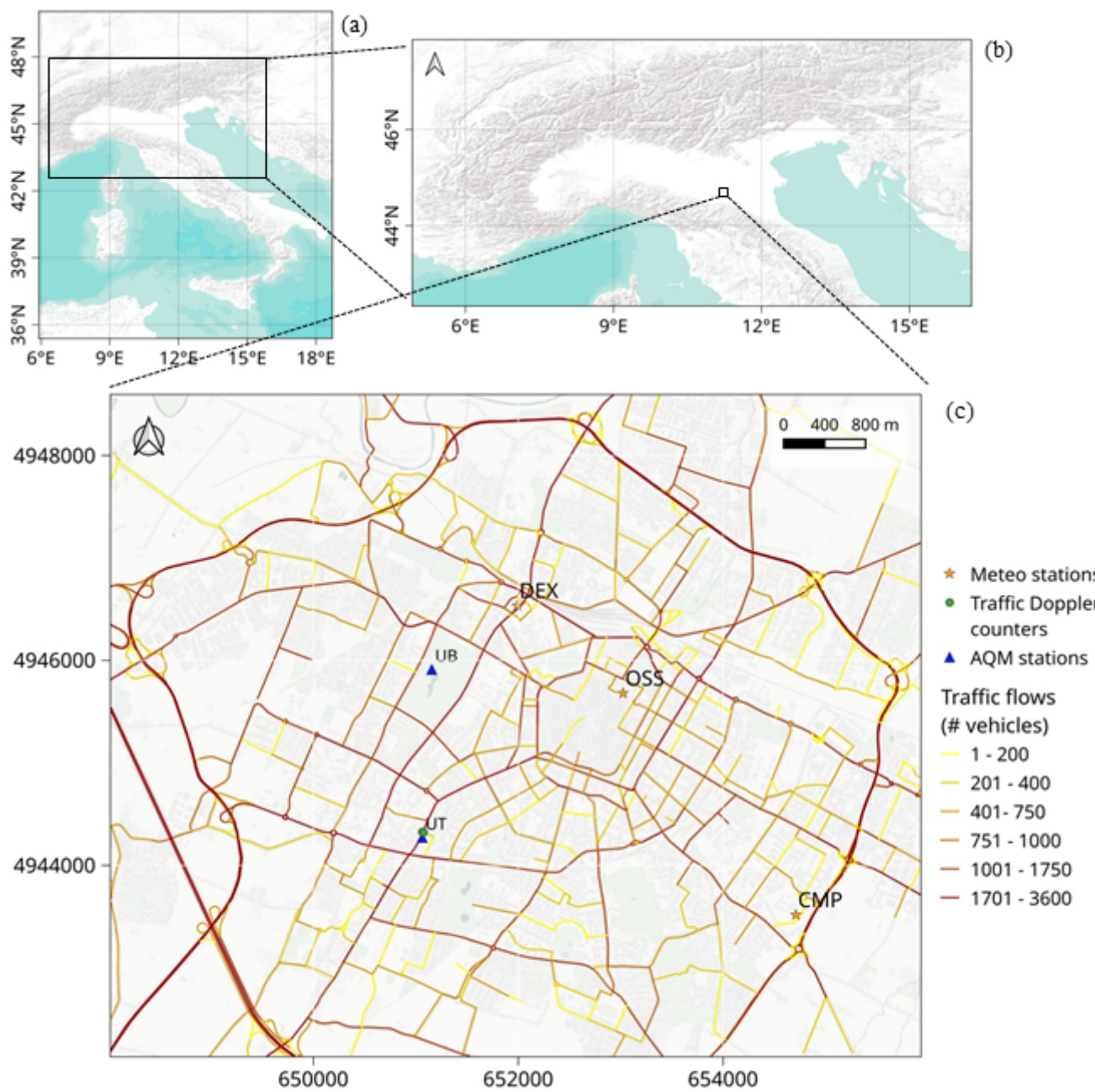

**Figure 3.** (a) Geographical context showing the location of the Po valley (from Esri, USGS, NOAA), (b) the location of Modena within the Po valley (from Esri, USGS, NOAA), and (c) an overview of the GRAL domain in Modena. Panel (c) shows simulated traffic flows generated by PTV VISM during the morning rush hour of a typical working day, along with marked positions for two urban air quality monitoring stations, urban meteorological stations and traffic radar Doppler counters used to adjust traffic modulation profiles.

## 3.1 Description of the integrated modelling approach

In this case study, the VERT emission model has been integrated into a comprehensive modelling suite specifically designed for the assessment of $NO_x$ concentrations over a domain covering most of the urban area of Modena with a very high horizontal resolution (4 m). While the system is used to assess the contribution of all the most important urban sources of $NO_x$, its design makes it particularly well suited to investigate the impact of the transport sector.

The main tools composing the integrated modelling approach are the following:

1. PTV VISUM, a macroscopic transport model designed to simulate traffic flows, taking into account factors such as road capacity, demand patterns and travel times (Heyken Soares et al., 2021). In our case study, this model was run by the municipality of Modena, using a predefined road network and including an estimate of the volume of trips between different origin-destination pairs (origin-destination matrix). The output of the model is the number of vehicles travelling on the reference road network for the morning rush hours (from 07:30 to 09:30 local time), divided into two reference categories, light vehicles (cars, mopeds and motorcycles) and heavy vehicles (lorries), and the corresponding average speed.

2. The VERT emissions model, which directly takes as input the traffic data provided by PTV VISUM, together with the local fleet composition and the road characteristics. This tool estimates the traffic emissions using the reference EF proposed by EMEP/EEA (Ntziachristos and Samaras, 2023; Ntziachristos and Boulter, 2023; Mellios and Ntziachristos, 2023).

3. The GRAMM-GRAL Lagrangian dispersion model. It is an advanced tool tailored to simulate the dispersion and deposition of pollutants in urban areas. It has been designed to take into account the presence of obstacles such as buildings, bridges and portals in the reconstruction of the flow field, and is particularly suited to provide a detailed understanding of the behaviour of pollutants in complex urban environments. A detailed description of GRAMM-GRAL can be found in Oettl (2015a, b, c, 2021); Oettl and Veratti (2021); Oettl and Reifeltshammer (2023).

## 3.2 Set-up of the integrated modelling approach

In order to implement a comprehensive modelling approach for the city of Modena, we collected and processed various input datasets. To characterise road traffic conditions, we integrated traffic flow estimates from the PTV VISUM model with historical traffic counts from induction loop spires at key intersections. In addition, radar Doppler counts, collected during winter of 2016 near the traffic air quality station of the city, complemented this data (see Ghermandi et al. (2020) for further details). The synergy among these datasets enabled a thorough analysis of the traffic situation, providing spatially distributed information and tailored traffic modulations. An overview of the road traffic volumes for the morning rush hours (between 07:30 and 09:30 local time) together with the location of radar Doppler sensors used for tailored modulation is provided in Fig. 3 panel (c).

Traffic emissions were calculated using VERT, incorporating traffic flows and speeds from PTV VISUM and data on the local fleet composition. The latter was derived from the national vehicle register (ACI, 2023) and then normalised by actual kilome-

tres travelled for each vehicle category, as estimated by the Italian Institute for Environmental Protection and Research (ISPRA, 2023). Supplementary data, such as the average air temperature and an estimate of the mean trip distance travelled by urban

vehicles, were obtained respectively from the meteorological reference station of the city and from the urban mobility plan (PUMS, 2023). The average temperature during the simulation period was recorded as 9.5°C, while the average trip distance was set at 2.5 km. The emission computations with the VERT package were performed on a 20-Core Intel Xeon Gold 6230 2.10GHz cluster node and took approximately 920 s of wall clock time.

$NO_x$ emissions from domestic heating, industry, waste treatment and other mobile machinery were also included in the simu-

lation. Estimates of these sources were taken from the regional emission inventory (INEMAR, 2019) and spatially distributed to different areas of the city. Emissions from agricultural machinery (other mobile sources) were represented as diffuse sources and allocated to rural areas, while the remaining emissions were integrated as point sources and distributed using different proxy variables such as building characteristics and land use classification. Further description on the methodology used to spatially distribute urban emissions can be found in Veratti et al. (2021), while Fig. A1 in the appendix shows the daily modu-

lation profiles derived from the traffic measurements and daily modulation profiles used for other emission sectors.

The GRAMM-GRAL model was set-up over two nested domains centred in the city of Modena. The outer domain, with an extension of 30 km x 30 km and a resolution of 200 m, was reserved for the Eulerian non-hydrostatic model GRAMM. This model solves the conservative equations for momentum, enthalpy, mass and humidity to reconstruct the large-scale wind field conditions, taking into account the contrasts in land use and the corresponding surface fluxes of heat, momentum and humidity. It uses

only local meteorological measurements and soil parameters, without requiring external initial and boundary conditions from large-scale models to drive the simulations. Topography and land use data were obtained from Geoportale-Emilia-Romagna (2023) and the Corine Land Cover database, updated to 2018 (CCL, 2018). Hourly meteorological observations of temperature, wind speed and direction were provided by three meteorological stations, CMP, DEX and OSS, located at altitudes of 10 m, 40 m and 50 m above ground level, respectively, as shown in Fig. 3 panel (c). Large scale wind patterns reconstructed by GRAMM

were used as boundary conditions for GRAL, run at the city scale over a domain of 7.9 km x 6.5 km (Fig. 3, panel c), with a horizontal resolution of 4 m. GRAL firstly reconstructed the urban wind speed and direction, taking into account the presence of urban obstacles, and secondly performed the Lagrangian dispersion of the pollutant sources provided as input.

To represent transport from sources outside the area of interest, concentrations measured at a rural background station, 40 km north of Modena, were used. This station is influenced only by long-range transport and is not affected by direct local sources

(Ghermandi et al., 2020). While acknowledging the possibility of small horizontal and vertical gradients in the background concentrations, this approach is considered reliable in view of the considerable homogeneity of concentrations and meteorological variables observed within the Po valley (Pernigotti et al., 2012; Scotto et al., 2021; Squizzato et al., 2013). Consequently, the rural background concentrations were added to the modelled urban concentrations to obtain the final concentrations.

The simulation period spans from 8 January to 8 March 2020, before the strict lockdown restrictions were imposed in northern

Italy for the COVID pandemic. The computation was performed at hourly time steps through the transient dispersion mode, which was chosen to ensure a more accurate representation of the concentration fields compared to the steady-state option,

albeit at a higher computational cost. See Oettl (2015a, b, c) for further details about possible GRAL configurations.

## 3.3    Results from the application of the integrated modelling approach

Hourly $NO_x$ concentrations simulated by the model were evaluated at two urban air quality monitoring stations in Modena. One station is located within a public park on the western side of the historical city centre, representing urban background conditions, while the other is along a busy urban road near a major intersection, where traffic is expected to be the primary source of pollution. Fig. 3 panel (c) depicts the precise locations of the two monitoring stations within the study area.

Fig. 4 compares the daily averaged $NO_x$ concentrations simulated by the modelling system with the observed values at the two

reference sites. To extend the insights from the urban modelling tools, the figure also incorporates observed $NO_x$ concentrations from the rural background station, which is intended to represent the contribution of emission sources outside Modena.

The comparison shows a generally good agreement between simulated and observed concentrations, particularly at the traffic site, where the simulated hourly average of $128 \pm 106\ \mu\mathrm{g\,m}^{-3}$ closely aligns with the observed average of $112 \pm 89\ \mu\mathrm{g\,m}^{-3}$. This agreement is further reflected by a low Mean Bias (MB), equal to $-13\ \mu\mathrm{g\,m}^{-3}$ and corresponding to -10% of Normalised

Mean Bias (NMB). Moreover, the Pearson correlation coefficient of 0.72 highlights a strong positive correlation between modelled and measured values. On the other hand, at the urban background site where the influence of traffic emissions on the overall concentration diminishes, the model's performance generally tends to decrease. This is particularly evident on January 9, 10, 14, and 23, when specific meteorological conditions (wind speeds below 2 $m\ s^{-1}$ and recurrent thermal inversions) favoured pollutant accumulation. Under these conditions, the model struggles to reproduce the observed signal particularly

at the urban background site. Here, modelled average concentrations are $52 \pm 37\ \mu\mathrm{g\,m}^{-3}$ while the observed average is $95 \pm 83\ \mu\mathrm{g\,m}^{-3}$. This discrepancy is reflected in a MB of $-39\ \mu\mathrm{g\,m}^{-3}$, corresponding to -42% of NMB, although associated to a satisfactory Pearson correlation coefficient of 0.62. Apart from the influence of meteorological factors, potential sources of uncertainty may lie in the estimation of non-traffic emission sources, such as domestic heating and industrial combustion, which characterise the area surrounding the urban background monitoring station. Less detailed estimation methods are used

for these sources and local proxies may not fully represent anthropogenic activity at this location. This highlights the need for improved estimation methods for non-traffic emission sources to improve the overall performance of the model, a task that falls beyond the scope of this study.

To complement the statistical analysis, the ability of the modelling system to reproduced the observed trend is also assessed using a set of indicators recommended by Hanna and Chang (2012) for the evaluation of urban dispersion models, including

FAC2, NAD, NMSE and FB, defined as reported in the Appendix. These benchmarks, which aim to ensure acceptable model performance, can be summarised as follows:

    – FAC2 > 0.30: At least 30% of the predicted concentrations should fall within a factor of two of the observed values.

    – NAD < 0.50: The fractional error area should be less than 50%.

    – |FB| < 0.67: The relative mean bias should be less than a factor of 2.

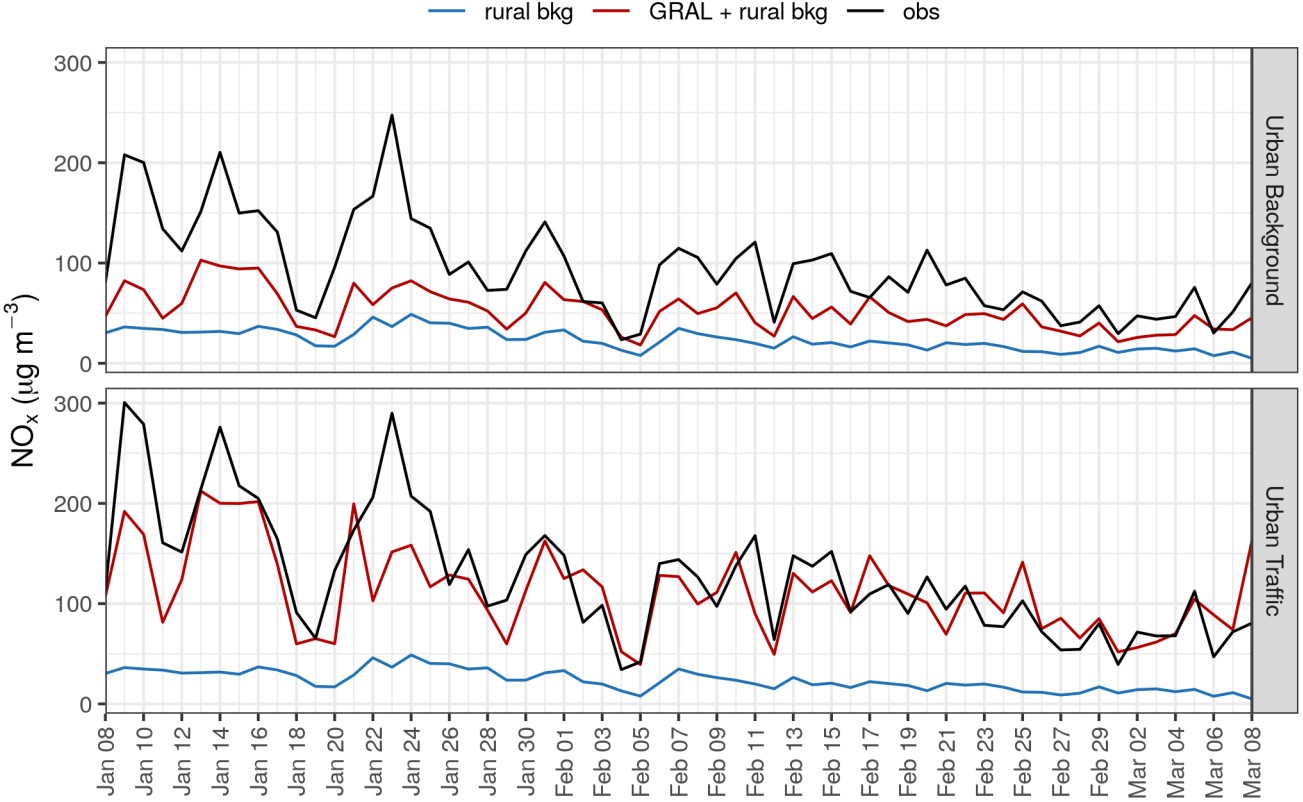

**Figure 4.** Daily time series showing observed and simulated $NO_x$ concentrations at urban traffic and background sites from 8 January to 8 March 2020, together with daily measured concentrations at the rural background station. Note that the simulated concentrations include the rural background contribution.

– NMSE <6: The random scatter should be less than 2.4 times the mean.

     While improvements in the estimation of non-traffic sources would further enhance model performance, the integrated modelling system consistently meets the acceptance criteria at both stations (Table 1). This highlights the ability of the models to capture the spatial and temporal variations in $NO_x$ concentrations, indicating its potential for accurate air quality modelling in urban environments. The same results also underline the effectiveness of VERT, coupled with detailed traffic information,

in quantifying traffic-related emissions in the urban environment.

     The second part of the assessment evaluates the modelled diurnal cycles compared to the measured values. This is important to determine the ability of a particular model to accurately represent urban daily maxima and the diurnal variation of predicted concentrations throughout the day. Fig. 5 shows the comparison of modelled and observed $NO_x$ daily mean cycles, along with their corresponding 25th and 75th percentiles, at both urban traffic and urban background stations. In addition, the diurnal trend

of $NO_x$ concentrations measured at the rural site is shown on the same plot to complement the information provided at the

**Table 1.** Model performance statistics of hourly $NO_x$ concentrations computed for the period 8 January to 8 March 2020 at the two urban air quality monitoring stations.

| Station | MB ($\mu g\,m^{-3}$) | NMB (%) | FAC2 | NAD | FB | NMSE | r |
|---|---|---|---|---|---|---|---|
| Urban background | -39 | -42 | 0.63 | 0.26 | 0.53 | 1.07 | 0.62 |
| Urban Traffic | -13 | -10 | 0.80 | 0.05 | 0.11 | 0.37 | 0.72 |

urban scale.

At the traffic site, modelled and observed $NO_x$ concentrations are generally very well aligned, and the two diurnal peaks are effectively captured by the modelling system. In contrast, at the urban background station, the model systematically underestimates the observed cycle, especially during the morning and evening peaks. This underestimation may be due to the inability

of the model to accurately represent the $NO_x$ sources in the area surrounding the urban background station. For example, the location of wood-burning stoves in the city and the estimation of the emissions associated with solid fuels (pellet, wood, etc.) remain highly uncertain.

In this simulation, the burning of wood for domestic heating was mainly attributed to rural areas, whereas some of these emissions may actually occur in more central locations of the city and contribute to local pollution, even during night time, as

noted by Bigi et al. (2023). In addition, $NO_x$ emissions from domestic heating due to the combustion of compressed natural gas (CNG) have been spatially distributed using the volume of each building as a spatial proxy, which may not accurately reflect the actual distribution. All of these factors may contribute to the underestimation observed at the urban background site, particularly in the early morning and late evening when domestic heating activity is at its highest.

Although some limitations have been identified, the integrated modelling approach demonstrates its value as a tool for assessing

traffic emissions at the kerbside. This strength opens up the possibility of using VERT, in combination with a high-resolution dispersion model, to assess different traffic emission scenarios, including changes in the fleet composition, the introduction of low emission zones and variations in traffic flows.

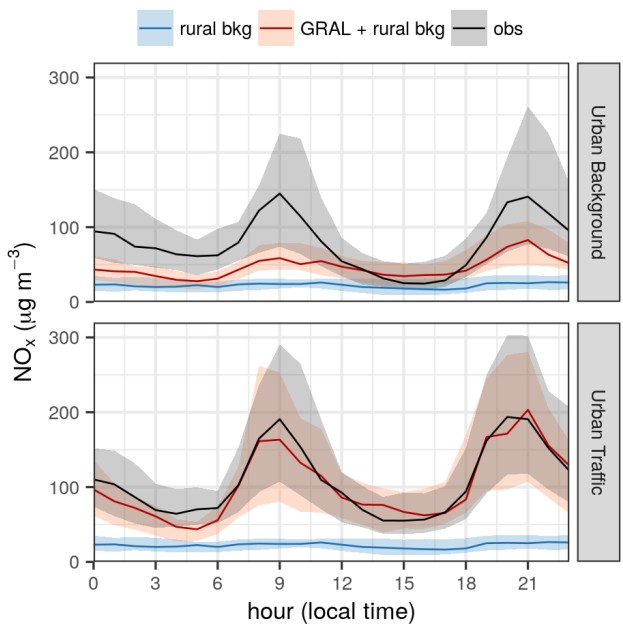

**Figure 5.** Mean daily cycle of observed $NO_x$ concentrations at urban stations (black), at the rural background station (blue) and modelled by GRAMM-GRAL plus the rural background contribution (red). The solid lines represent the daily mean cycle, while the shaded area shows the variability between the 25th and 75th percentiles.

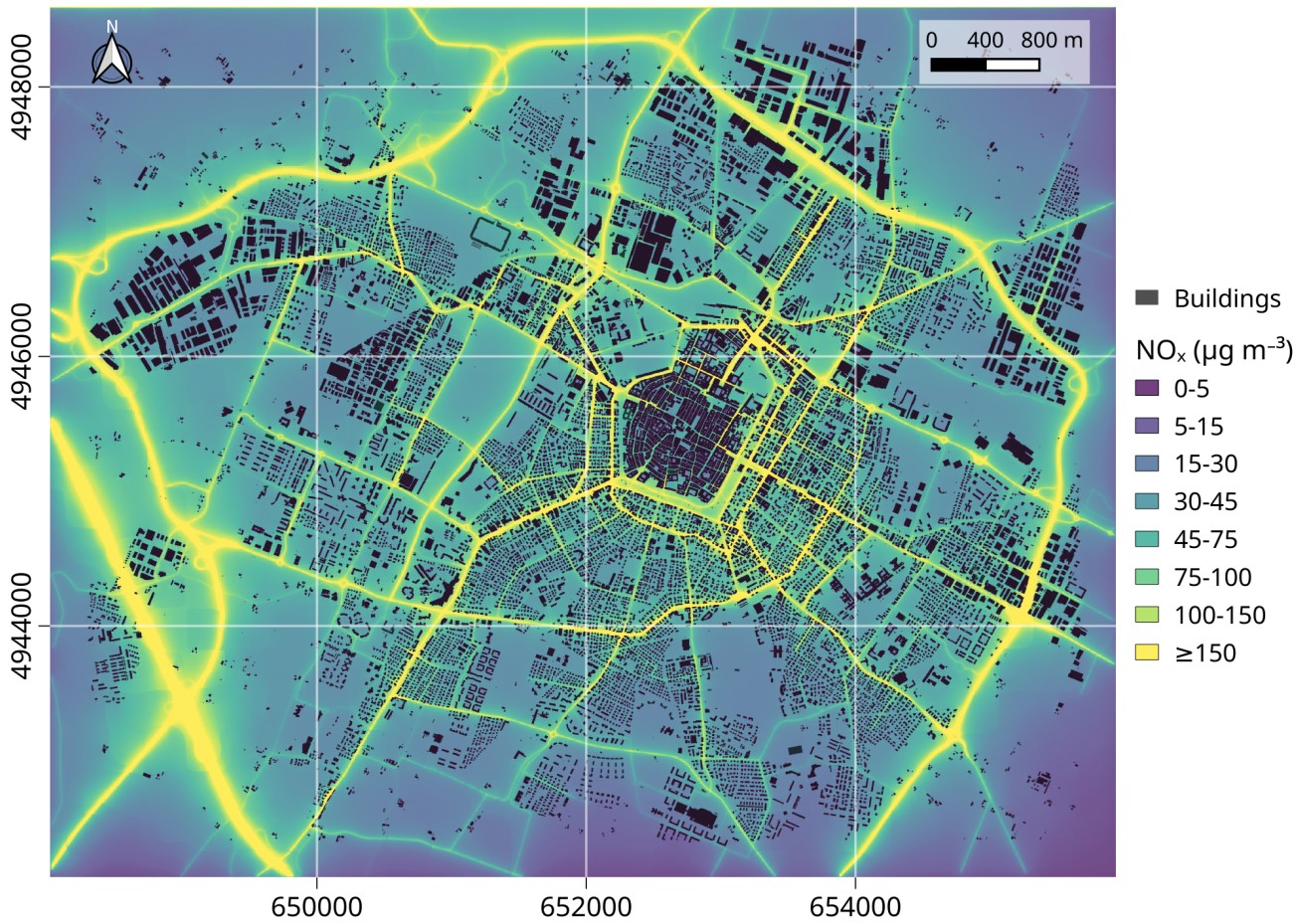

**Figure 6.** Spatial distribution of average simulated $NO_x$ concentrations in the city of Modena from January 8 to March 8, 2020, as modelled by GRAMM-GRAL. Urban building locations are depicted in black, while $NO_x$ concentrations are colour-coded from purple to yellow, illustrating variations across the city.

Fig. 6 presents the spatial distribution of $NO_x$ concentrations simulated using the integrated modelling approach. The map clearly shows the concentration gradient along major roads, with particularly high levels along the urban ring road around the city centre and the motorway in the lower left corner. Additionally, concentration peaks are also found in more central urban areas characterised by dense traffic and elevated building density, which trap pollutants and contribute to local hotspots.

# 4 Case study 2: Application and validation of VERT at the regional scale

The second case study focuses on the use of VERT to assess transport emissions on a larger scale, encompassing the entire Emilia-Romagna region, a large area in the Po valley of approximately 22,000 square kilometres. The main objective of this section is to quantify transport emissions using traffic estimates provided by the regional authority, and to compare the results obtained from VERT with estimates derived from the reference emission inventory for the same region (INEMAR, 2019). This application sets the stage for investigating the performance of VERT and to provide insights about its applicability for

estimating transport emissions on a regional scale.

## 4.1 Methods

Since 2001, the Emilia-Romagna region has been using a transport modelling tool to support its extra-urban mobility system, covering both private and public transport modes and their potential integration. The PTV VISUM software serves as

the reference modelling tool for these simulations, providing estimates for the inter-zonal movements within the region and interactions with neighboring areas and regional crossings. The full range of mobility possibilities are allocated to different potential destinations using a comprehensive socio-economic dataset, including population, employment and student data, divided into zones. Using origin-destination matrices and local traffic measurements, the PTV VISUM model assesses vehicle traffic patterns on a reference road network of approximately 7,000 arcs and 2,500 nodes during the typical morning rush hour

of a working day (from 7:00 to 9:00 local time). Vehicles are also categorised into four different groups: cars, light commercial vehicles, heavy commercial vehicles and mopeds/motorcycles. Fig. 7 gives an overview of the traffic flows as estimated by PTV VISUM for the morning rush hours. Urban traffic flows for each municipality are not included in the simulation because the granularity required for accurate urban traffic patterns (complex intersections, varying speed limits, pedestrian interactions and more frequent stops) is usually beyond the scope of regional models. Urban traffic patterns require highly detailed data,

including traffic signals, pedestrian crossings, local road layouts and variations in daily and weekly traffic flows. Collecting and maintaining this level of detail for an entire region is complex and resource intensive. Potential sources for extending regional traffic flows to the city level include the urban mobility plans of medium and large cities, which can provide accurate and reliable data on traffic movements and vehicle speed distribution at a level of detail not achievable at the regional level.

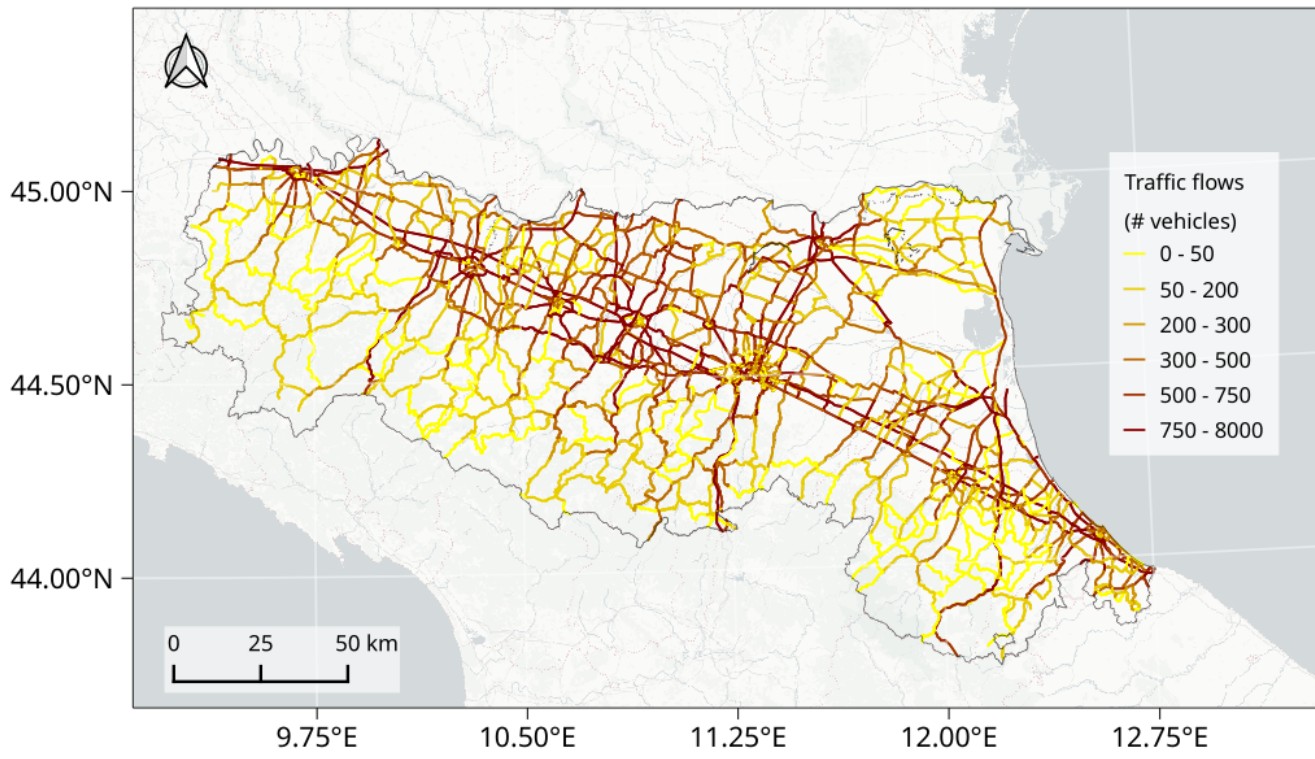

**Figure 7.** Traffic flows simulated by PTV VISUM during the morning rush hours (from 7:00 to 9:00 local time) for a typical working day in the Emilia-Romagna region (Po valley). Basemap from Esri, USGS, NOAA.

To comprehensively represent vehicle flows throughout the year, 72 distinct scenarios were devised from the estimate pro-
vided by PTV VISUM, each tailored to account for seasonal variations, weekday and weekend dynamics, and daily load patterns. In addition, the vehicle speed provided by the PTV VISUM model for the morning rush hours was adjusted to reflect off-peak scenarios by using measured flow curves for the reference network. This approach allowed the assignment of realistic vehicle speeds corresponding to the road's capacity.

In order to fully assess transport emissions for each of the 72 traffic scenarios, VERT was run using the reference road fluxes
and the estimated vehicle speeds. In addition, seasonal average temperature data for 2019, sourced from the ERA5 archive (Hersbach et al., 2018), were incorporated and averaged for the entire region to accurately estimate cold start and evaporative emissions. The fleet composition for 2019 was used to determine the percentage breakdown of EMEP/EEA classes required by VERT, categorised by fuel type, Euro emission standard, engine capacity and vehicle mass. Fleet information was extracted from the national vehicle register (ACI, 2023) and adjusted by actual mileage for each vehicle category to ensure data accuracy
(ISPRA, 2023). Emission estimates for extra-urban and motorway roads were performed on a 52-core Intel Xeon Gold 5320 2.20 GHz cluster node and took approximately 349,100 s.

Since the PTV VISUM simulations only cover extra-urban and motorway traffic, the total regional fuel consumption was used

to estimate the share of emissions due to urban traffic. The fuel consumption calculated by VERT for petrol, diesel, CNG and liquefied petroleum gas (LPG), based on PTV VISUM fluxes and velocities, was subtracted from the total regional fuel consumption for 2019 (MASE, 2019). This fuel difference was allocated to urban traffic and distributed among all municipalities, using population as a proxy variable. Then, in an iterative process, VERT was used to estimate the distance required for the urban vehicle fleet to consume the missing fuel, which in turn was used to calculate the urban transport emissions for each municipality. These procedures were performed on an Intel i7-5500U 2.40 GHz single-core laptop and took approximately 730 s.

INEMAR, the software used to compile the regional emission inventory, follows the EMEP/EEA methodology similar to VERT. However, differences include the fleet composition, the procedures for estimating urban traffic flows and speeds, the allocation of cold start emissions between extra-urban and urban traffic, and the formulation for calculating evaporative running losses. Table 2 summarises the main input data and methodologies used by both models. INEMAR uses fleet composition data from ACI (2023) for 2019, while VERT adjusts these data based on estimated kilometres travelled per vehicle class (ISPRA, 2023), reflecting the actual presence of vehicles on the road. VERT estimates traffic flows iteratively, whereas INEMAR uses empirical formulas to estimate the total annual kilometres travelled per vehicle category. It combines fuel consumption and traffic flows on extra-urban roads and motorways to derive total kilometres travelled, and then estimates urban traffic flows by calculating the difference between the total estimate and the extra-urban/motorway calculation. In addition, INEMAR uses data from various urban mobility plans for reference speeds, while VERT uses measured data from traffic campaigns in Modena, considered representative for other municipalities in the region. Finally, INEMAR assigns all cold start emissions to urban traffic for each municipality, while VERT distinguishes between extra-urban and urban cold start emissions.

### 4.2 Emission evaluation and comparison with the reference emission inventory

The total annual emissions calculated by VERT and INEMAR are presented in Table 3, categorised by road type (roadways, extra-urban, urban) and annual totals. Overall, VERT shows a good agreement with INEMAR for $NO_x$, PM exhaust, $SO_2$, PM wear and evaporative NMVOC emissions, with deviations ranging from -24% to 19% in terms of annual totals. This confirms the reliability of VERT and its ability to provide comparable estimates with the reference emission inventory. For other pollutants, such as CO, exhaust NMVOC, and $NH_3$, the difference between VERT and INEMAR is more pronounced, with absolute deviations of 49%, 76% and 38%, respectively. These discrepancies, particularly for CO and NMVOC, are mainly due to urban traffic (69% and 83% respectively), where emissions are calculated on the basis of fuel consumption and are therefore subject to greater uncertainty. Several factors, including differences in the fleet composition, in the urban vehicle speeds, fuel blends and associated calorific values, can lead to significant disparities between the two models. In this comparison, VERT is likely to attribute a higher proportion of kilometres travelled to gasoline vehicles in urban traffic conditions compared to INEMAR, resulting in higher NMVOC and CO estimates. Similarly, heterogeneities in fleet composition may also explain the discrepancy for $NH_3$ emissions.

Despite these factors, the disparities between VERT and INEMAR are of the same order of magnitude as those found in similar studies carried out in the Po valley. For example, Pallavidino et al. (2014), who compared the output of the traffic emission

**Table 2.** Comparison between VERT and INEMAR setup and calculation methods.

| Emission type | Input/Details | INEMAR | VERT |
|---|---|---|---|
| - | Fleet composition | ACI (2023) year 2019 | ACI (2023) year 2019 adjusted according to ISPRA (2023) |
| - | Extra-urban and motorway fluxes | PTV VISUM simulations | PTV VISUM simulations |
| - | Urban fluxes | Empirical formulas for average vehicle mileage and fuel consumption | Computed with an iterative process based only on fuel consumption |
| - | Extra-urban and motorway velocities | Assessed using measured speed-flow curves | Assessed using measured speed-flow curves |
| - | Urban velocities | Derived from various urban traffic plans | Derived from measured traffic campaigns in Modena |
| Exhaust | Methodology | Eq. (2), Eq. (3) and Eq. (4) | Eq. (2), Eq. (3) and Eq. (4) |
| | EFs | EMEP/EEA 2020 | EMEP/EEA 2020 |
| | $EF_i$dgr | Function of vehicles speed | Constant with vehicle speed |
| | Cold start emissions | Urban includes both extra-urban and urban | Divided between extra-urban and urban |
| Non-Exhaust | Methodology | Eq. (7) | Eq. (7) |
| | EFs | EMEP/EEA 2020 | EMEP/EEA 2020 |
| Evaporative running losses | Methodology | Eq. (B1) and Eq. (B2) | Eq. (11) |
| Resuspension | Methodology | Not included | Eq. (8) and custom EFs from Amato et al. (2012) |

model TREFIC with INEMAR for the province of Turin, found differences ranging from 3% to 92% for $NO_x$, CO, $PM_{10}$, NMVOC and $NH_3$, with larger gaps observed for NMVOC, as in the present case study. This highlights the significant uncertainties that still exist in the estimation of NMVOC emissions from transport sources and underlines the potential for different
methodologies to produce divergent results.

Other authors in Europe have made comparisons of traffic emission estimates between the reference local emissions inventory, typically compiled using a top-down approach, and tailored bottom-up methods. For instance, Chan et al. (2023) compared the output of Yeti with the reported 2015 emissions at the city level from the Berlin Senate inventory. Employing various Yeti configurations, the results showed disparities between the two approaches within the ranges of 11-20% for CO, 5-99% for
525 hydrocarbons, 4-48% for NOx and 2-49% for PM. This confirms that estimates related to volatile organic compounds are the ones affected by larger uncertainties, also in different areas of Europe.

Further comparisons between fine scale bottom-up approaches and European top-down inventories (EC4MACS, TNO MACC-II, and TNO MACC-III) were performed for seven urban areas in Norway (López-Aparicio et al., 2017). These investigations revealed that the three top-down regional inventories underestimated $NO_x$ and $PM_{10}$ traffic emissions by approximately 20-

80% and 50-90%, respectively. Other authors, such as Borge et al. (2012), conducted a comparison between two of the most widely used traffic emission methodologies in Europe, EMEP/EEA and HBEFA, in assessing traffic emissions for the city of Madrid. Their analysis showed that the annual totals for $NO_x$ from HBEFA were 21% higher than those from EMEP/EEA, while the differences for primary $NO_2$ were in the order of 13%.

These studies provide evidence that the discrepancies observed between VERT and INEMAR are consistent with similar comparisons made for other European cities and areas. In addition, the findings from the same studies highlight the importance of employing bottom-up methods alongside top-down approaches to achieve more accurate estimates of traffic emission, particularly for volatile organic compounds, which are crucial for air quality modelling and policy development.

Simulations with VERT were also carried out to estimate the resuspension of road dust caused by vehicle movement, whose emissions are not included in the local emission inventory. In the absence of a standardised method for assessing this component of $PM_{10}$ emissions, simulations were conducted using both the EPA-42 methodology and a second approach based on user defined EF. The EPA-42 methodology was applied using a silt load of $0.2\,\mathrm{g\,m^{-2}}$, while the second approach relied on different EF for cars, mopeds, light trucks and heavy trucks, which were set respectively to 12.5, 1.1, 45 and $250\,\mathrm{mg\,km^{-1}}$ per vehicle, as in the range proposed by Amato et al. (2012). The results revealed significant discrepancies between the two methods (Table 3), with the EPA-42 approach consistently overestimating the second approach. These relative differences are equal to 46% for motorways, 60% for non-urban roads and 51% for urban roads. Although similar results have been found in the past by other authors (Amato et al., 2016; Harrison et al., 2021; Casotti Rienda and Alves, 2021), it is important to note that the EPA-42 methodology is based on data collected from U.S. roads, while the selected emission factors come from studies carried out in Spain. As a result, their applicability to a situation different from the one in which they were developed may be subject to some uncertainty, and neither approach can be clearly considered as a reference for the Po valley. Nevertheless, these simulations provide a preliminary assessment of the potential emission contributions from road dust resuspension.

Fig. 8 illustrates the spatial distribution of annual emissions for $NO_x$ and $PM_{10}$ across the municipalities of the Emilia-Romagna region. To facilitate comparison between municipalities, total emissions are expressed in tons per square kilometre. Generally, municipalities with lower emissions are located in the southern part of the region, in hilly and mountainous areas, where traffic flows are lower and the municipalities are less populated than in other locations. On the other hand, areas with higher emissions are characterised by the presence of motorways, which contribute additional emissions from both urban and rural networks. This is particularly evident for $NO_x$ (Fig. 8, panel a), where emissions from motorways are exacerbated by the higher speeds compared to rural and urban driving. In contrast, motorway emissions are less pronounced for $PM_{10}$ (Fig. 8, panel b), as the non-exhaust component dominates PM emissions, with the latter being less significant on motorways because braking and cornering are more frequent in urban and rural driving.

**Table 3.** Comparative analysis between VERT simulations and INEMAR estimates for the Emilia-Romagna region. The Table includes the annual totals and the percentage difference between the two methods.

| Pollutant | motorway | | | extra-urban | | | urban | | | Totals | | |
|---|---|---|---|---|---|---|---|---|---|---|---|---|
| | VERT (ton) | INEMAR (ton) | diff (%) | VERT (ton) | INEMAR (ton) | diff (%) | VERT (ton) | INEMAR (ton) | diff (%) | VERT (ton) | INEMAR (ton) | diff (%) |
| $NO_x$ | 19649 | 13065 | 34 | 12748 | 12722 | 0 | 7209 | 8025 | -11 | 39606 | 33812 | 15 |
| CO | 7428 | 9443 | -27 | 6371 | 5357 | 16 | 38498 | 12019 | 69 | 52297 | 26819 | 49 |
| PM exhaust | 373 | 229 | 39 | 254 | 223 | 12 | 186 | 204 | -09 | 813 | 656 | 19 |
| $NMVOC_{exh}$ | 856 | 501 | 41 | 1112 | 635 | 43 | 9582 | 1658 | 83 | 11549 | 2794 | 76 |
| $NMVOC_{evap}$ | 42 | 55 | -24 | 41 | 59 | -31 | 1198 | 1577 | -24 | 1281 | 1691 | -24 |
| $SO_2$ | 27 | 21 | 22 | 23 | 23 | 1 | 14 | 13 | 6 | 64 | 57 | 11 |
| $NH_3$ | 131 | 232 | -77 | 91 | 170 | -86 | 132 | 87 | 34 | 354 | 489 | -38 |
| wear TSP | 868 | 760 | 12 | 724 | 1029 | -42 | 473 | 422 | 11 | 2064 | 2211 | -7 |
| wear $PM_{10}$ | 543 | 458 | 16 | 486 | 685 | -41 | 319 | 287 | 10 | 1348 | 1430 | -6 |
| wear $PM_{2.5}$ | 293 | 259 | 12 | 256 | 365 | -43 | 169 | 150 | 11 | 718 | 774 | -8 |
| resusp $PM_{10}$ EPA-42 | 2015 | - | - | 1687 | - | - | 1804 | - | - | 5506 | - | - |
| resusp $PM_{10}$ custom EF | 1088 | - | - | 682 | - | - | 879 | - | - | 2649 | - | - |

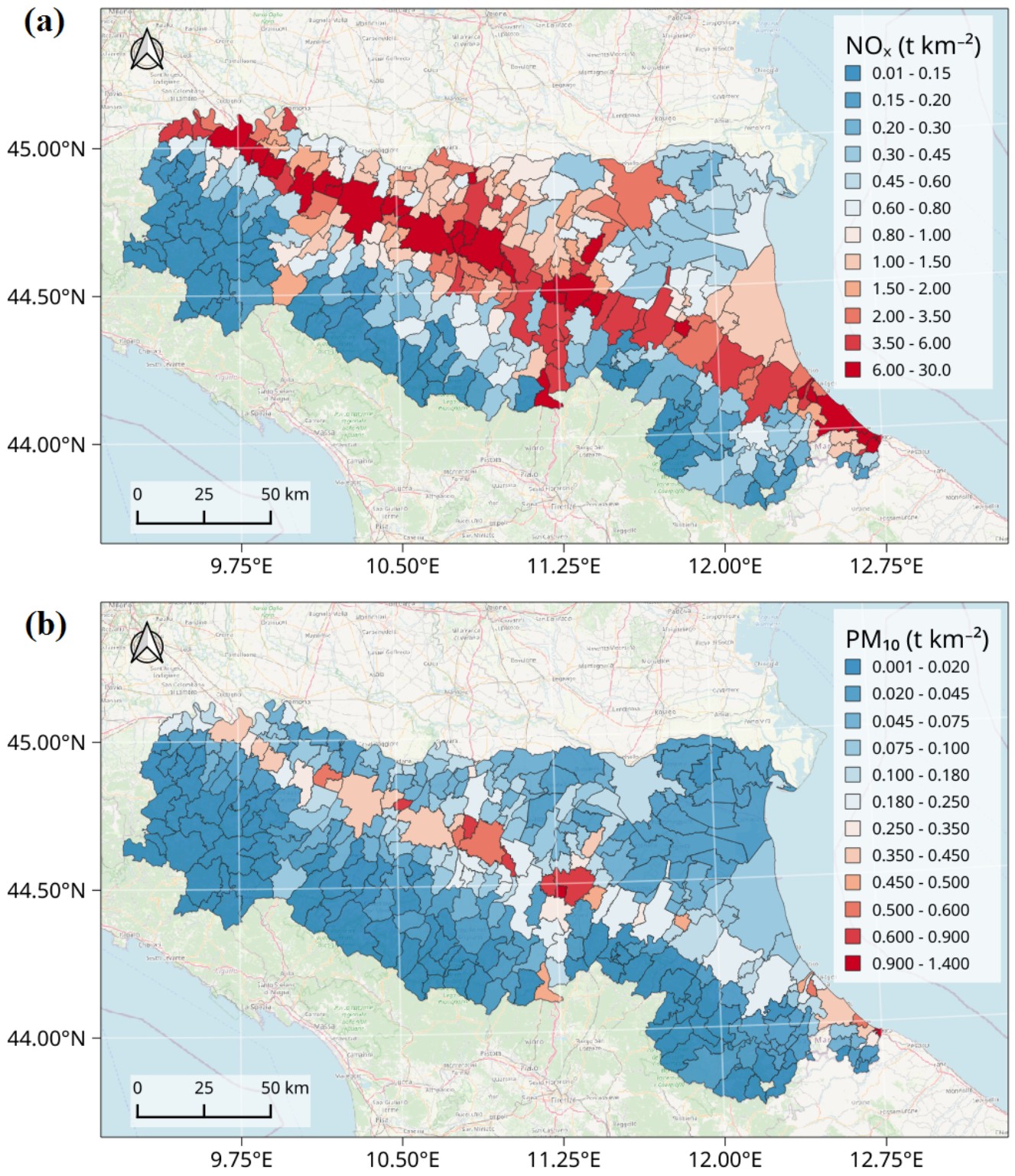

**Figure 8.** Spatial distribution of $NO_x$ (panel a) and $PM_{10}$ (panel b) emissions simulated by VERT for each municipality in the Emilia-Romagna region (from © OpenStreetMap contributors 2023). The cumulative emissions account for contributions from urban, rural, and motorway roads.

## 5 Conclusions

This study presents VERT, a bottom-up traffic emissions model implemented in the R programming language. VERT is capable of estimating emissions for a wide range of pollutants and greenhouse gases starting from traffic estimates along a reference road network, accompanied by data on vehicle fleet composition and fuel blends. Compared to existing tools in the literature, VERT is characterised by simplicity of operation and rapid configuration, even for users with limited programming experience. At the same time, the tool offers remarkable flexibility in user input accommodating three different types of vehicle flows. VERT also includes emission factors for different emission calculations, such as hot exhaust, cold start, evaporative, non-exhaust and resuspension, whose implementation follows the methodology proposed by the EMEP/EEA and the 2006 IPCC guidelines.

VERT was integrated into a modelling framework together with PTV VISUM and GRAMM-GRAL to evaluate its ability to accurately estimate traffic emissions in a real case study. This integrated approach allows for the validation of VERT in simulating $NO_x$ emissions in the town of Modena, an urban hotspot of the Po valley. VERT emissions were fed to the Lagrangian dispersion suite GRAMM-GRAL to simulate $NO_x$ concentrations, which in turn are compared with observations from two urban air quality monitoring stations, one located in an area representative of urban background conditions and the other representative of traffic conditions. The results show that the integrated modelling approach effectively reproduces the observed trends, especially at the traffic site where the associated emissions are expected to be the major contributors, confirming the ability of VERT to provide reliable estimates of traffic emissions. Although the accuracy of the modelling system at the urban background site is lower than at the urban traffic site, its performance at both sites meets the acceptance criteria defined in the literature for urban dispersion modelling.

The effectiveness of VERT in reproducing traffic emissions is further evaluated at a regional scale, in a domain covering the entire Emilia-Romagna region, located south of the Po valley. VERT simulations are performed for 72 different traffic scenarios using measured traffic counts and simulations of the PTV VISUM model, calibrated for the morning rush hours (from 7:00 to 9:00 local time) of a typical working day. The results of VERT are then extrapolated to annual totals and compared with INEMAR, the reference emission inventory for the same region. VERT demonstrates strong agreement with INEMAR, especially for $NO_x$, PM exhaust, $SO_2$, PM wear and evaporative NMVOC, with differences ranging from -24% to 19%. For other pollutants, such as CO, NMVOC and $NH_3$, the discrepancy increases up to 76%, but is still within the range (3-92%) of similar comparisons carried out in other regions of the Po valley. At the same time, these results highlight the persistent uncertainty associated with the estimation of NMVOC emissions from traffic.

Simulations with VERT are also performed to account for the resuspension of road dust at the regional scale. Employing both the EPA-42 methodology and vehicle-specific emission factors, a comprehensive range of potential contributions of resuspension to $PM_{10}$ is provided in the latest section of the paper.

In conclusion, VERT is a versatile and user-friendly bottom-up traffic emissions model that effectively estimates traffic emissions at both urban and regional scales. Its ability to simulate emission patterns and its alignment with reference emission inventories make it a valuable tool for air quality modelling and emission reduction strategies.

*Code availability.* The source code for VERT used in this study can be accessed via the digital object identifier https://doi.org/10.5281/zenodo.12549513 under the GNU GPL-3 license. The Git repository for VERT is also available at https://gitlab.com/GiorgioVeratti/vert. The official software code for GRAMM-GRAL is available at https://gral.tugraz.at/ and through the Git repository at https://github.com/GralDispersionModel. The specific version of the GRAMM-GRAL code used in this study is also available via the following permanent link: https://doi.org/10.5281/zenodo.10728500.

## 600 Appendix A: Model evaluation

To assess the performance of the model in reproducing $NO_x$ concentrations, several statistical indicators were employed. These indicators were derived using the following notation:

M: Modelled values
O: Observed values
n: Number of model-observation pairs

Average modelled value:

$$610 \quad \bar{M} = \frac{1}{n} \sum_{i=1}^{n} M_i$$

Average observed value:

$$\bar{O} = \frac{1}{n} \sum_{i=1}^{n} O_i$$

The following metrics were used for evaluation:

$$615 \quad MB = \frac{1}{n} \sum_{i=1}^{n} (M_i - O_i)$$

$$NMB = \frac{1}{n} \sum_{i=1}^{n} \frac{(M_i - O_i)}{O_i}$$

$$r = \frac{\sum_{i=1}^{n} (M_i - \bar{M})(O_i - \bar{O})}{\sqrt{\sum_{i=1}^{n} (M_i - \bar{M})^2 \sum_{i=1}^{n} (O_i - \bar{O})^2}}$$

$$FAC2 = \text{Fraction of data where } 0.5 \leq \frac{M_i}{O_i} \leq 2$$

$$NMSE = \frac{\overline{(O-M)^2}}{\bar{O} \cdot \bar{M}}$$

$$FB = \frac{\overline{O-M}}{0.5 \cdot (\bar{O} + \bar{M})}$$

$$NAD = \frac{\overline{|O-M|}}{(\bar{O} + \bar{M})}$$

$$RMSE = \sqrt{\frac{1}{n} \sum_{i=1}^{n} (M_i - O_i)^2}$$

## Appendix B: Evaporative emissions formulation inclded in the INEMAR emission model

The INEMAR emission model accounts only for evaporative running losses. The formulation included in the model is represented by Eq. (B1):

$$E_i^k run_{\text{inemar}} = n.veh^k \cdot L^k \cdot EF_{\text{hr}} \cdot \delta \cdot 10^{-6} \tag{B1}$$

$EF_{\text{hr}}$ is given by Eq. (B2), $n.veh^k$ and $L^k$ are respectively the number of vehicles travelling on the road segment $k$ and the length of the road segment $k$ itself. $\delta$ takes the values 1 for gasoline-powered vehicles, 0 for vehicles powered by other fuels, 0.2 for mopeds, and 0.4 for motorcycles.

$$EF_{\text{hr}} = 0.136 \cdot \exp(-5.967 + 0.04259 \cdot \text{RVP} + 0.1773 \cdot T) \cdot \epsilon \tag{B2}$$

*RVP* represents the fuel vapor pressure in kPa, *T* is the mean air temperature for the period of interest and $\epsilon$ equals 1 for vehicles without a canister fuel system and 0.1 for vehicles equipped with a canister.

## Appendix C: Emission modulations

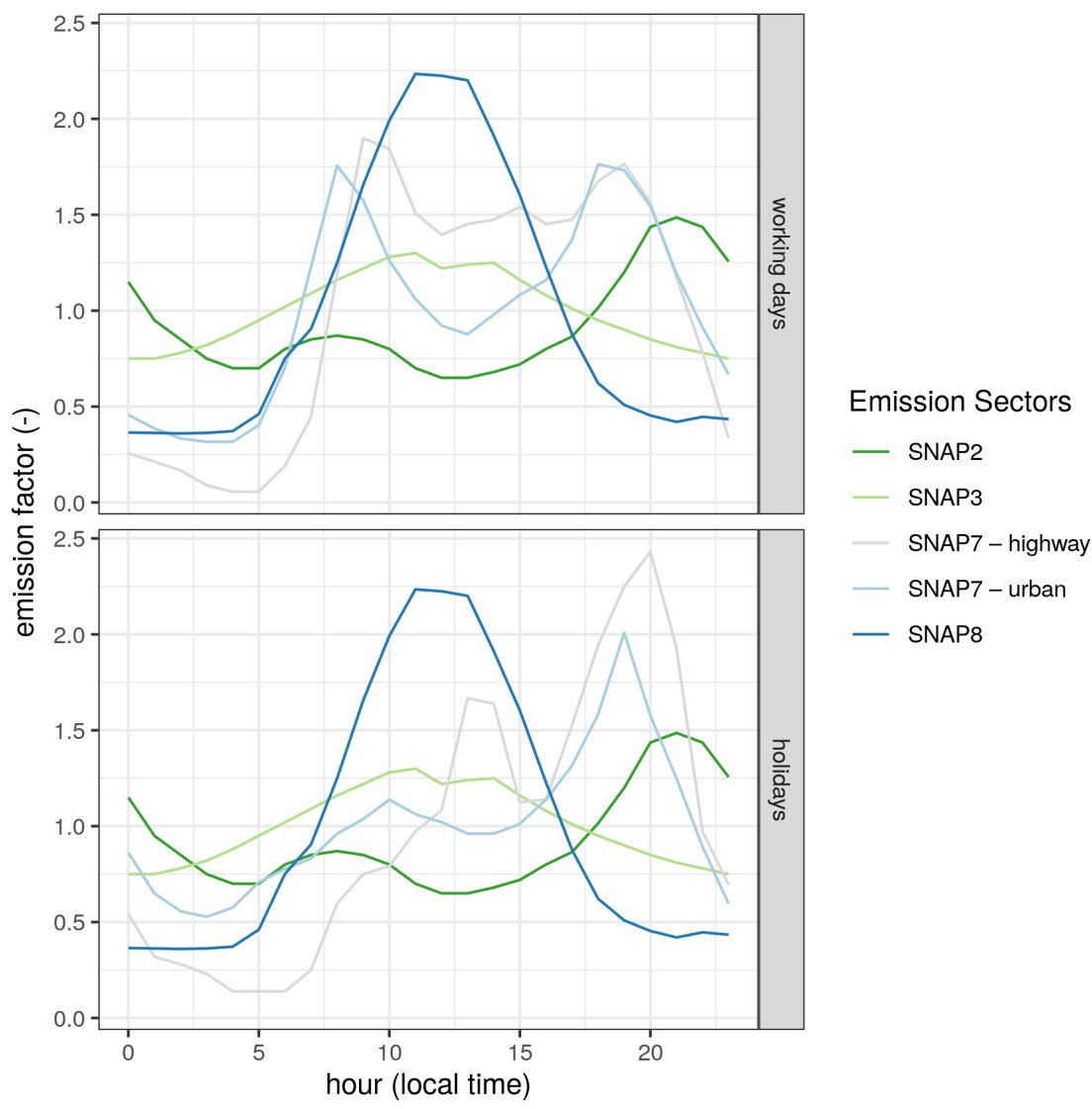

**Figure A1.** Emission modulations used for GRAL simulations.

*Author contributions.* GV designed and developed the VERT tool, performed the VERT and GRAMM-GRAL simulations, and wrote the original draft of the paper. AB provided scientific input to the development of VERT and reviewed and edited the manuscript. ST and GG provided essential resources and contributed to data interpretation. All co-authors provided feedback and contributed to the manuscript.

*Competing interests.* The authors declare that they have no conflict of interest.

*Acknowledgements.* This research has received partial funding from the "Ecosystem for sustainable transition in Emilia Romagna (ECOSIS-TER)" project, identified with code ECS_00000033, funded by the European Union NextGenerationEU programme—Call for tender n. 3277 dated 30 December 2021, Award Number: 0001052 dated 23 June 2022—under the National Recovery and Resilience Plan (NRRP) Mission 4, Component 2, Investment Line 1.5: "Establishing and strengthening" of "innovation ecosystems for sustainability", building "territorial leaders of R&D". The authors would like to thank the Municipality of Modena for generously sharing the traffic simulation data of the city and Arpae Emilia-Romagna for providing data and insights on the traffic measurement campaign. Special thanks go to Chiara Agostini, Simona Maccaferri and Michele Stortini of Arpae Emilia-Romagna for their valuable contributions and insightful feedback on the description of the INEMAR emission model and its functionalities. Their commitment to sharing insights on the interpretation of results has enriched the quality of this work.

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
