# Peer review of "Description and validation of VERT 1.0, an R-based framework for estimating road transport emissions from traffic flows"

_EGUsphere, 2024_

## Author Comment (AC2)

We would like to thank both reviewers for their time and valuable comments, which greatly improved the manuscript. Our responses to the reviewers' comments are given below. Please note that references to specific lines in the manuscript correspond to the author's track changes file.

Anonymous Referee #1, 22 Apr 2024

The paper presents VERT, an R package to estimate traffic emissions following the EMEP/EEA methodology. The article is well organized showing some practical applications of the package. In general, I would like to suggest also to include specific section or comments regarding the foreseen calculation time and efforts also in relation to the resolution of the estimates.

We acknowledge the reviewer's point that the article lacked a dedicated section discussing the computational cost of the emissions calculations performed in the two test cases, as well as an analysis of the scalability of the code across different numbers of cores and machines. To address this, we have added a new section, 2.6 "Computational Performance" (lines 273-291). In this section we have assessed the performance of VERT using four different machines with different numbers of cores available, tested on a reference road network of 500 roads. In addition, we have provided details of the computational costs of the two case studies in the main text on lines 350-351, lines 485-486, and 493-494.

Referring the line number of the preprint, please find here some specific comments:

Line 41 – the non-exhaust and resuspension emissions are treated in VERT. I suggest adding specific comments also in this abstract on traffic emissions.

This comment was not entirely clear to us. We were unsure whether the reviewer was suggesting an explicit mention in the abstract of VERT's ability to calculate non-exhaust and resuspension emissions, or whether additional background information should be provided around line 41 in the Introduction. To address both possibilities, we have added a detailed description of VERT's capabilities in the Abstract (lines 4-5) and provided additional background information on non-exhaust traffic emissions in the Introduction on lines 44-48.

Line 136 – please add in text or refer how the vehicle speed is calculated.

The VERT model is not designed to estimate traffic flows and related velocities. Instead, its purpose is to compute emission estimates using input traffic data from external sources, such as microscopic or macroscopic traffic models, or measurements from various devices, including Doppler radar counters, automated cameras, or induction loop spires. When only traffic counts are available for a given road, several empirical equations from the literature can estimate vehicle velocity based on peak hour traffic flow and the road's vehicle capacity. These estimated velocities can then be used as input for the VERT model. This clarification has been added to the main text at lines 157-162.

Line 435 – The authors try to explain the calculation differences between VERT and similar software:

"Differences can be identified in the aggregation of vehicle classes of the fleet composition and the way this data is used in the calculation. INEMAR uses the total number of registered vehicles in the region (ACI, 2023) and assumes that these vehicles travel on the reference road network according to the flows simulated by PTV VISUM. Emissions are then calculated on the basis of vehicle 440 movements. Conversely, VERT uses a percentage fleet composition to estimate an average EF for each vehicle category of the simulated flows. The reference EF are then multiplied by the

simulated flows to obtain the final emissions. In addition, the fleet composition processed by INEMAR is not normalised by actual kilometres travelled by vehicle category."

These comments are not clear. To support the analysis, it will be useful reporting a synoptic table resuming all the main hypothesis between the two model on all the input variables, also specifying e.g. if the road network, main fluxes and fleet composition are the same and how they are treated by the models.

We acknowledge that a table resuming all the main features and input data to both the models can help the reader to identify the differences of the two softwares and methodologies. To enhance clarity, the text mentioned by the reviewer was rephrased and Table 2 has been added to the main text. See lines 495-513.

It will be useful also adding some comments on the road network resolution; are all the streets treated by the model? If not, is it possible to propose how estimating emissions on all the roads?

As reported in the text (lines 487-488), the resolution of the available traffic data covers only extra-urban roads and motorways; urban flows are not included in the simulation. Figure 6 shows the spatial distribution of the road network included in the simulation. A possible source of additional information for estimating urban traffic flows is the urban mobility plans of medium and large cities, which can provide valuable data for estimating traffic fluxes and vehicle velocities. Comments on this have been added to the main text (lines 467-473). However, it is important to note that the VERT model is not designed to simulate or represent traffic flows for a given situation but rather to compute emissions when and where traffic flows are available. To complement the use of VERT, the Tier 1 and Tier 2 approaches described in Ntziachristos and Samaras (2023) can be used in cases where low-level data are available for urban scales.

During the revision of the main manuscript, we identified and corrected a number of additional errors, which are summarized below:

- Figure 1 has been revised to include parking lots in addition to roads (see the right box under the output section). Additionally, the blue bubble indicating meteorological input has been added to the wear_emis.R and PM_resusp.R functions to reflect their dependence on meteorology.

- There were typographical errors in Table 1, which have now been corrected. Specifically, lines 1 and 2 were swapped for some statistics.

- Reference Update: The reference for Bigi et al. (2023) has been updated from "under discussion" to "published". Now the published version is the following:
  Bigi, A., Veratti, G., Andrews, E., Collaud Coen, M., Guerrieri, L., Bernardoni, V., Massabò, D., Ferrero, L., Teggi, S., and Ghermandi, G.: Aerosol absorption using in situ filter-based photometers and ground-based sun photometry in the Po Valley urban atmosphere, Atmospheric Chemistry and Physics, 23, 14 841–14 869, https://doi.org/10.5194/acp-23-14841-2023, publisher: Copernicus GmbH, 2023.

- The apex $k$ has been added to the terms $n.\text{veh}$ and $L$ in Eq. 11.

Reference:

Ntziachristos, L. and Samaras, Z.: 1.A.3.b.i-iv Road transport 2019 — European Environment Agency, https://www.eea.europa.eu/publications/emep-eea-guidebook-2019/part-b-sectoral-guidance-chapters/1-energy/1-a-combustion/1-a-3-b-i/view, 2023.

---

## Author Comment (AC3)

We would like to thank both reviewers for their time and valuable comments, which greatly improved the manuscript. Our responses to the reviewers' comments are given below. Please note that references to specific lines in the manuscript correspond to the author's track changes file.

Larysa Pysarenko, 08 Jun 2024

\Paper entitled "Description and validation of VERT 1.0, an R-based framework for estimating road transport emissions from traffic flows", describes VERT (Vehicular Emissions from Road Traffic) created in R package, that allows estimating traffic emissions. This topic is essential for improving emission inventories for different spatial and temporal resolutions. In general, minor revision is recommended.

General comments:

1) Line 134 and Eq. 3. The authors mention experimental coefficients. What are the values of these coefficients? Are they incorporated into the VERT code? It would be better to provide more details on the procedure for their estimation. Were these coefficients approximated by a polynomial trend or somehow else?

The requested details have been added to the main text at lines 141-147. As stated, the values of these coefficients depend on the vehicle type, fuel, emission standard, engine size, road characteristics, and heavy duty vehicle load. They were estimated through regression analysis, resulting in a polynomial curve that fits the observed data. For more information on the procedure used for their determination, please refer to Kouridis et al. (2010) and Ntziachristos and Samaras (2023). These experimental coefficients are incorporated into VERT via dedicated data frames, such as *'vert::EF\$hot_ef_ldv_emep.eea.2020'*,*'vert::EF\$hot_ef_ldv_emep.eea.2023'*, *vert::EF\$hot_ef_hdv_emep.eea.2020'*, and *'vert::EF\$hot_ef_hdv_emep.eea.2023'*. These data frames can be inspected, modified, or updated by users who wish to test their own experimental coefficients.

Also, there are some hesitations concerning the term "D/velocity" in Eq.3. Taking into account that VERT can be used for simulating hourly emissions, do authors consider the possibility of almost zero vehicle speeds? It relatively frequently occurs in megapolises during rush hours or under some emergency conditions on the roads.

The experimental coefficients mentioned above are considered reliable by the EMEP/EEA methodology within a speed range of approximately 5-20 km h$^{-1}$ to 100-140 km h$^{-1}$, depending on vehicle category. It is recognised that in traffic jams or very congested situations, particularly in large cities, vehicles may operate at very low speeds. To better represent emissions under such conditions, a correction factor has been introduced into VERT to account for increased emissions at very low speeds. Specifically, when the vehicle speed falls below the threshold of the validity range of the proposed coefficients, the time spent on the road is increased by a factor of *w*, calculated as reported in the main text (lines 148-155). However, it is important to highlight that the model is tailored to driving scenarios and therefore idling emissions may not be accurately estimated.

2) Line 314-320. The authors mentioned the complex topography of the studied domain. It will be useful to provide additional details on land cover and topography data input to the model. Also, there is no information about meteorological input, including its temporal and spatial resolution.

The requested details about land cover, topography, and meteorological inputs have been added to the manuscript at lines 361-369. Specifically, the land cover data were sourced from the Corine Land Cover database (CCL, 2018), and topography data were obtained from the Geoportale Emilia-Romagna (2023). Meteorological input data, including hourly observations, were provided by three stations (CMP, DEX, and OSS) located at altitudes of 10, 40, and 50 meters above ground level. Figure 3 panel (c) has also been updated.

During the revision of the main manuscript, we identified and corrected a number of additional errors, which are summarized below:

- Figure 1 has been revised to include parking lots in addition to roads (see the right box under the output section). Additionally, the blue bubble indicating meteorological input has been added to the wear_emis.R and PM_resusp.R functions to reflect their dependence on meteorology.

- There were typographical errors in Table 1, which have now been corrected. Specifically, lines 1 and 2 were swapped for some statistics.

- Reference Update: The reference for Bigi et al. (2023) has been updated from "under discussion" to "published". Now the published version is the following:
  Bigi, A., Veratti, G., Andrews, E., Collaud Coen, M., Guerrieri, L., Bernardoni, V., Massabò, D., Ferrero, L., Teggi, S., and Ghermandi, G.: Aerosol absorption using in situ filter-based photometers and ground-based sun photometry in the Po Valley urban atmosphere, Atmospheric Chemistry and Physics, 23, 14 841–14 869, https://doi.org/10.5194/acp-23-14841-2023, publisher: Copernicus GmbH, 2023.

- The apex $k$ has been added to the terms $n$.veh and $L$ in Eq. 11.

References:

CCL: CORINE Land Cover, https://land.copernicus.eu/en/products/corine-land-cover, 2018.

Geoportale-Emilia-Romagna: Servizi cartografici regionali, https://geoportale.regione.emilia-romagna.it, 2023

Kouridis, C., Gkatzoflias, D., Kioutsoukis, I., Ntziachristos, L., Pastorello, C., and Dilara, P.: Uncertainty Estimates and Guidance for Road Transport Emission Calculations, https://doi.org/10.2788/78236, iSBN: 9789279153075 ISSN: 1018-5593, 2010.

Ntziachristos, L. and Samaras, Z.: 1.A.3.b.i-iv Road transport 2019 — European Environment Agency, https://www.eea.europa.eu/publications/emep-eea-guidebook-2019/part-b-sectoral-guidance-chapters/1-energy/1-a-combustion/1-a-3-b-i/view, 2023.